# Deforming Videos to Masks: Flow Matching for Referring Video Segmentation

**Zanyi Wang**[2,1*]   **Dengyang Jiang**[3,1*]   **Liuzhuozheng Li**[4,1*]   **Sizhe Dang**[1]   **Chengzu Li**[5]
**Harry Yang**[3]   **Guang Dai**[1]   **Mengmeng Wang**[6,1†]   **Jingdong Wang**[7‡]

[1]SGIT AI Lab, State Grid Corporation of China   [2]University of California, San Diego
[3]The Hong Kong University of Science and Technology   [4]The University of Tokyo
[5]University of Cambridge   [6]Zhejiang University of Technology   [7]Baidu

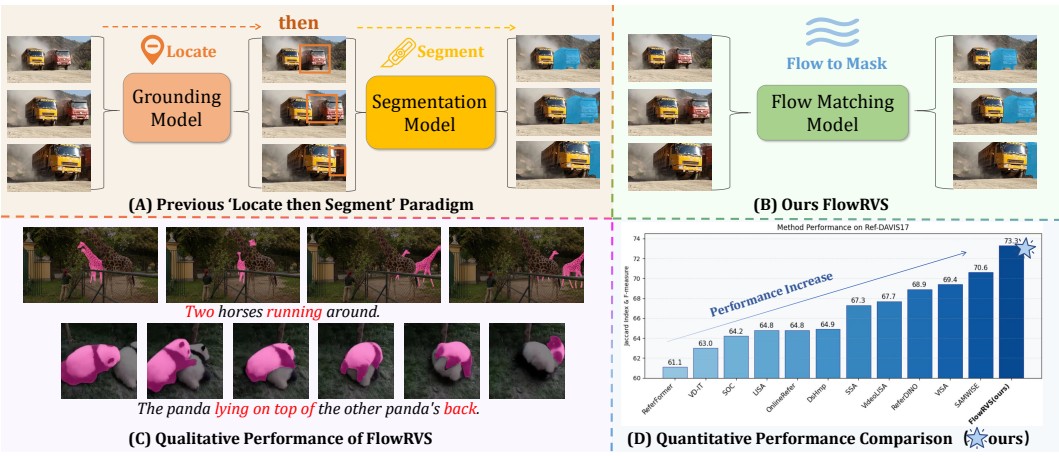

Figure 1: FlowRVS replaces the cascaded 'locate-then-segment' paradigm (A) with a unified, end-to-end flow model (B). This new paradigm avoids information bottlenecks, enabling superior handling of complex language and dynamic video (C) and achieving state-of-the-art performance (D).

## Abstract

Referring Video Object Segmentation (RVOS) requires segmenting specific objects in a video guided by a natural language description. The core challenge of RVOS is to anchor abstract linguistic concepts onto a specific set of pixels and continuously segment them through the complex dynamics of a video. Faced with this difficulty, prior work has often decomposed the task into a pragmatic 'locate-then-segment' pipeline. However, this cascaded design creates an information bottleneck by simplifying semantics into coarse geometric prompts (e.g., point), and struggles to maintain temporal consistency as the segmenting process is often decoupled from the initial language grounding. To overcome these fundamental limitations, we propose FlowRVS, a novel framework that reconceptualizes RVOS as a conditional continuous flow problem. This allows us to harness the inherent strengths of pretrained T2V models, fine-grained pixel control, text-video semantic alignment, and temporal coherence. Instead of conventional generating from noise to mask or directly predicting mask, we reformulate the task by learning a direct, language-guided deformation from a video's holistic representation to its target mask. Our one-stage, generative approach achieves new state-of-the-art results across all major RVOS benchmarks. Specifically, achieving a $\mathcal{J}\&\mathcal{F}$ of 51.1 in MeViS (+1.6 over prior SOTA) and 73.3 in the zero shot Ref-DAVIS17 (+2.7), demonstrating the significant potential of modeling video understanding tasks as continuous deformation processes. *Code is available at:* *https://github.com/xmz111/FlowRVS*

---

*Equal contribution. This work was completed during the internship at SGIT AI Lab.
†Corresponding author.   ‡Project lead.

# 1 INTRODUCTION

Referring Video Object Segmentation (RVOS) (Khoreva et al., 2018; Gavrilyuk et al., 2018; Hu et al., 2016) requires the machine to segment objects described by natural language queries, which is critical to intelligent systems to precept and interact with the real world (Jiang et al., 2025; Li et al., 2023). The core challenge of RVOS lies in resolving a fundamental spatio-temporal correspondence dilemma: anchoring abstract linguistic concepts onto a dynamic and fine-grained pixel space. Current paradigms often rely on instance-centric approaches, which first identify and then track object instances. While effective, even modern query-based (e.g., ReferFormer (Wu et al., 2022)) or VLM-based (e.g., LISA (Lai et al., 2024)) methods can introduce an information bottleneck by collapsing rich semantics into intermediate object-centric representations. This can limit holistic scene understanding and temporal consistency, especially as the segmentation of each frame, while conditioned, doesn't stem from a single, unified spatio-temporal deformation process (Liang et al., 2025b; Ren et al., 2024; Bai et al., 2024; Lin et al., 2025).

To address these limitations, we argue that pretrained Text-to-Video (T2V) models offer a fundamental solution since their native capabilities for fine-grained, text-to-pixel synthesis and spatio-temporal reasoning directly counter the bottlenecks of the 'locate-then-segment' paradigm. While some attempts use T2V models as powerful frozen feature extractors for a separate decoder (e.g., VD-IT (Zhu et al., 2024), HCD (Zhang et al., 2025)), this two-stage design decouples the model's generative dynamics from the final task. Our work fundamentally differs: we propose to repurpose the entire generative process itself, learning a direct, language-guided deformation flow from video to mask. DepthFM (Gui et al., 2025) have adapted the entire generative process itself for visual-to-visual tasks like depth estimation based on a image generation model. While these pioneering efforts validate the generative approach, they also expose a shared, critical blind spot: they fail to fully utilize the dynamic, text-driven reasoning that T2V models are capable of and RVOS demands. The feature-extraction approach remains decoupled, forcing a separate decoder to reconstruct temporal relationships from temporally isolated features, squandering the T2V model's inherent video coherence. Meanwhile, the image-to-depth flows proposed by previous styles completely neglect text condition, rendering them fundamentally incapable of addressing the core RVOS challenge: producing different masks for the same video based on varying textual queries. Thus, we argue that a deeper, more principled alignment with the T2V paradigm is required: one that treats the entire process as a single, unified, language-guided flow from video pixels to required masks.

This unified flow with existing powerful T2V pretrain model (e.g, Wan) brings several benefits: (1) their pixel-level synthesis training provides fine-grained control, which enables them to distinguish and handle more delicate objects when locating specific targets; (2) their text-condition generation ensures powerful multi-modal alignment, which allows them to ground rich linguistic semantics directly in the pixel space without as much information loss as first mapping in coarse geometric intermediaries; (3) their video-native architecture provides inherent spatio-temporal reasoning, naturally unifying language guidance with temporal consistency.

However, simply leveraging the T2V framework is not enough. As shown in Figure 3, standard T2V generation is a divergent process: it maps a simple noise prior to a set of possible videos, exploring a broad trajectory space. RVOS, conversely, is a convergent task: it must map a complex, high-entropy video to a single, low-entropy mask. This transforms the problem into a deterministic, guided information contraction, where the text query acts as the crucial selector that isolates the precise target from the rich visual input (e.g., distinguishing "the smaller monkey" from "the bigger monkey"). This core insight that RVOS is a convergent flow directly informs our contributions. To successfully manage this asymmetric transformation, we introduce a suite of principled adaptations: (1) a boundary-biased sampling strategy to force the model to master the crucial, high-certainty start of the trajectory where the video's influence is strongest; (2) a direct video injection mechanism to preserve the rich source context throughout the contraction process; and (3) a task-specific VAE adaptation and start point augmentation to create a stable latent space for this unique mapping.

Summarizing, our contributions are as follows:

- We reformulate RVOS as learning a continuous, text-conditioned flow that deforms a video's spatio-temporal representation into its target mask, directly resolving the correspondence between language and dynamic visual data.

- We propose a suite of principled techniques that successfully enable the transfer of powerful text-to-video generative models to this challenging video understanding task.

- Our proposed framework, FlowRVS, establishes a new state of the art on key benchmarks. Notably, it achieves a significant improvement of **1.6%** $\mathcal{J}\&\mathcal{F}$ on the challenging MeViS dataset and **2.7%** $\mathcal{J}\&\mathcal{F}$ on the zero-shot DAVIS 2017 benchmark.

## 2 RELATED WORK

**Referring Video Object Segmentation** aims to segment a target object within a video based on a natural language expression (Xu et al., 2018; Ding et al., 2023). This task demands both visual-linguistic understanding and robust temporal segmenting. Early approaches often adapted frame-level referring image segmentation models and appended temporal linking mechanisms as a post-processing step (Khoreva et al., 2018). More recent and competitive methods have evolved into more integrated yet predominantly multi-stage pipelines. A dominant paradigm involves a "locate-then-segment" strategy, where a powerful multi-modal model first grounds the textual reference to a spatial region, which then guides a separate segmentation process for each frame.

**"Locate-then-Segment" Paradigm** manifests in several forms. A significant breakthrough came with the introduction of query-based architectures, inspired by the success of DETR-style (Zhu et al., 2020) transformers in vision tasks (Wu et al., 2022; Yan et al., 2024b), this new paradigm reframed RVOS by treating language as a query to the visual features. Furthermore, multimodal model based approaches like LISA (Lai et al., 2024), VISA (Yan et al., 2024a) and ReferDINO (Liang et al., 2025b;a) leverage a pretrained model's reasoning or grounding ability, such as LLaVA(Liu et al., 2023) or DETR-based GroundingDINO (Ren et al., 2024), to perform initial object localization, and then introduce a custom-designed mask decoder to generate the final segmentation. A similar philosophy is seen in the "VLM+SAM" family of methods (Luo et al., 2023; Wu et al., 2023), which use a Vision-Language Model for bounding box prediction, followed by a generic segmentation model like SAM (Kirillov et al., 2023; Cuttano et al., 2025) to produce pixel-level masks (He & Ding, 2024; Lin et al., 2025; Pan et al., 2025). Another line of work explores repurposing generative models: VD-IT (Zhu et al., 2024) first extracts features from a pretrained text-to-video diffusion model and then feeds these features into a separate DETR-like architecture for mask prediction. While these methods have pushed the performance boundaries, their reliance on intermediate representations—whether object queries or extracted features—can introduce information bottlenecks that prevent a truly holistic, end-to-end optimization of the video-to-mask correspondence problem.

**Generative Modeling** is largely catalyzed by the advent of latent diffusion models (Rombach et al., 2022). Building on this success, the frontier rapidly expanded into the temporal domain, leading to a surge of powerful text-to-video (T2V) models (Liu et al., 2024; Wan et al., 2025; Gao et al., 2025). Recent works have begun to leverage these models for RVOS. A notable approach (e.g., VD-IT (Zhu et al., 2024), HCD (Zhang et al., 2025)) utilizes T2V models as powerful frozen feature extractors for a separate segmentation decoder. Concurrent work ReferEverything (REM) (Bagchi et al., 2025) also explores video-to-mask generation but relies on direct one-step prediction. Our work fundamentally differs: instead of **extracting features** or **learning a static mapping**, we repurpose the entire generative process, fine-tuning the core model to **learn a direct video-to-mask deformation flow**. This avoids the bottleneck inherent in a two-stage pipeline. Furthermore, unlike conditional generation frameworks like ControlNet(Zhang et al., 2023) that add external guidance to a divergent, noise-to-image process, our method learns a convergent, discriminative transformation from the video source itself. Beyond diffusion, Flow Matching (Lipman et al., 2022; Liu et al., 2022) offers a significant theoretical advancement by learning a velocity field to transport samples along a deterministic ODE path. This has been leveraged for visual tasks like depth estimation (e.g., DepthFM (Gui et al., 2025)), but these methods are typically text-agnostic. Our work makes a critical distinction: we introduce the natural language query as the core conditional force that modulates the entire ODE path. This elevates the framework from a simple translation to a dynamic, multi-modal reasoning engine, reframing RVOS as a learned, conditional deformation.

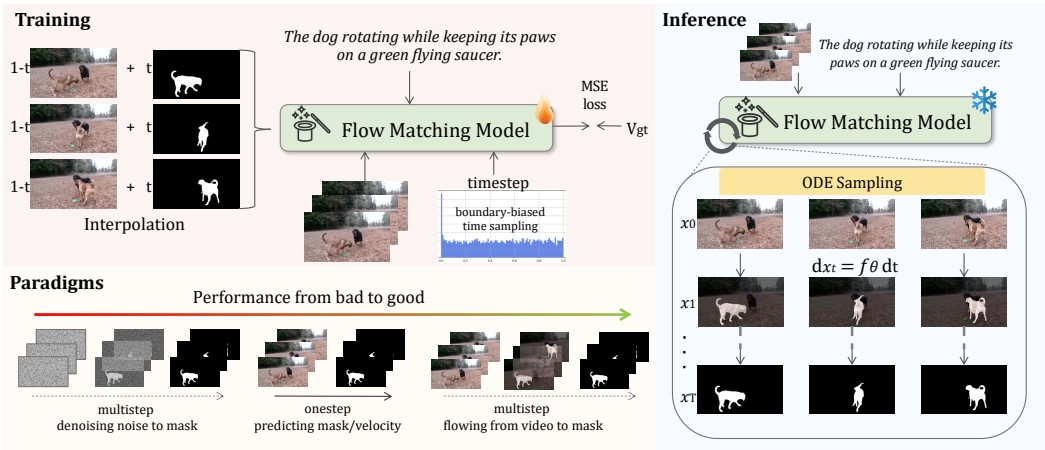

Figure 2: FlowRVS reformulates RVOS as a text-conditioned continuous flow, learning a velocity field via Flow Matching stabilized by boundary-biased time sampling in latent space. During inference, an ODE solver uses this field to deterministically deform the video latent to the target mask, this video to mask paradigm superior to noise-based or one-step prediction approaches.

## 3 METHOD

### 3.1 PROBLEM REFORMULATION: RVOS AS A CONTINUOUS FLOW

Traditionally, RVOS is approached as a discriminative, one-step prediction task. A model $f_\theta$ is trained to learn a direct mapping $M = f_\theta(V, c)$ from a video-text pair to a mask sequence. However, this direct mapping is fundamentally challenged by the need to collapse a dynamic, high-dimensional video into a precise pixel-mask under the complex constraints of a linguistic instruction, all within a single transformation.

To overcome this limitation, we depart from direct prediction and reconceptualize RVOS as a **text-conditioned continuous flow problem**. We propose to model segmentation as a gradual, deterministic deformation process that transforms the video's representation into the target mask's. This is governed by an Ordinary Differential Equation (ODE), where our goal is to learn the velocity field $\mathbf{v}(\mathbf{z}_t, c, t)$ that dictates the evolution of a latent state $\mathbf{z}_t$:

$$\frac{d\mathbf{z}_t}{dt} = \mathbf{v}(\mathbf{z}_t, c, t), \quad \text{with boundary conditions } \mathbf{z}_0 \sim \mathcal{P}_{video} \text{ and } \mathbf{z}_1 \sim \mathcal{P}_{mask}. \quad (1)$$

The trajectory starts from the video latent $\mathbf{z}_0$ and is guided by the text query $c$ to terminate at the specific mask latent $\mathbf{z}_1$. This transforms the learning objective from mastering a single, complex global function to learning a simpler, local velocity field.

However, adapting this generative-native paradigm to a discriminative task like RVOS is not straightforward, but a fundamental inversion of the generative process, as illustrated in Figure 3. Standard T2V generation is a divergent, one-to-many process: it starts from a simple, fixed noise distribution and has a broader exploration space in the initial steps to generate a diverse set of plausible videos. In contrast, our approach is a convergent, video-text-to-one task. It begins with a complex, high-entropy video latent $\mathbf{z}_0$ and must follow a more tightly mapped direction to a single, correct mask. Here, the text query $c$ is no longer a creative prompt but a critical, disambiguating force. The initial velocity computed from $\mathbf{z}_0$ must be precise enough to distinguish "the smaller monkey" from "the bigger monkey." An error in this first step is irrecoverable, dooming the entire trajectory to fail. This places paramount importance on correctly learning the starting point of the flow.

### 3.2 TRANSFERRING TEXT-TO-VIDEO MODEL TO RVOS

A naive, uniform treatment of the trajectory, inherited from generative modeling, fails to account for the unique asymmetric nature of the video-to-mask flow. This asymmetry—a high-certainty,

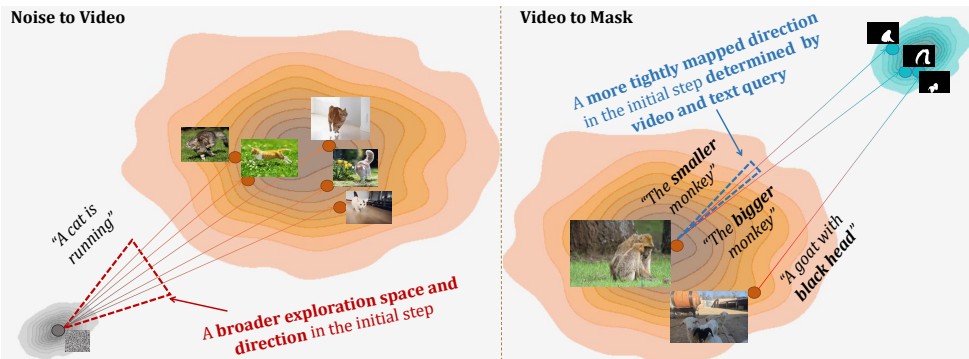

Figure 3: Repurposing a generative process for a discriminative task. Unlike standard T2V generation which maps noise to diverse videos (left), our method maps a complex video to a single mask (right). This transforms the process into a deterministic, convergent task where the text query is the crucial element that selects the precise target from the visual input (e.g., distinguishing the 'smaller' from the 'bigger' monkey).

structured start and a low-certainty, sparse end—demands a non-uniform approach to learning the velocity field. Therefore, we introduce a suite of three synergistic strategies grounded in a single principle: fortifying the flow's origin. These are designed not as independent tweaks, but as a cohesive framework to successfully adapt the powerful T2V model for RVOS.

**Boundary-Biased Sampling (BBS).**  We hypothesize that the most critical learning signal resides at the beginning of the trajectory, where the model computes the initial "push" away from the video manifold based on the text query. To exploit this, we introduce BBS, a curriculum learning strategy that oversamples timestep $t = 0$. By concentrating the gradient updates on this initial, high-influence decision, we force the model to first master the crucial text-guided velocity computation. As empirically demonstrated in Table 2, this focused learning strategy is the key to stabilizing the training process, transforming the failing baseline into a highly effective model by ensuring a well-posed initial value problem for the ODE.

**Start-Point Augmentation (SPA).**  To prevent the model from overfitting to discrete points on the data manifold and to encourage the learning of a smoother, more generalizable flow, we introduce Start-Point Augmentation (SPA). During training, we transform the initial video latent $\mathbf{z}_0$ through a stochastic encoding and normalization process. This technique effectively presents the model with a richer, locally continuous distribution of starting points centered around the original video latent. This acts as a powerful regularizer, forcing the model to learn a velocity field that is robust not just on the manifold, but also in its immediate vicinity.

**Direct Video Injection (DVI).**  In our video-to-mask formulation, the initial video latent $\mathbf{z}_0$ is not merely a starting point, but the foundational context for the entire transformation. To ensure this context remains accessible throughout the flow, we introduce Direct Video Injection (DVI). We implement this by concatenating the original video latent $\mathbf{z}_0$ with the current state $\mathbf{z}_t$ along the channel dimension at each ODE step without introducing heavy computational burden. This transforms the velocity prediction at every subsequent point from $v(\mathbf{z}_t, t)$ to $v([\mathbf{z}_t, \mathbf{z}_0], t)$, explicitly conditioning each local update on the global origin. This simple yet effective strategy provides a persistent, high-fidelity reference to the source video, preventing trajectory drift and improving fine-grained accuracy with negligible computational overhead.

### 3.3   ANALYSIS OF ALTERNATIVE PARADIGMS

To motivate our final choices, we first analyze the fundamental limitations of three plausible alternative paradigms for adapting a T2V model to RVOS. As empirically validated in our ablation studies, each of these alternatives fails due to a core mismatch with the nature of RVOS. To ensure a fair comparison, all alternative paradigms were built upon the same Wan2.1 pre-trained model and

trained under the exact same supervised setting (same optimizer, learning rate, and duration) as our final model. And we implemented the same fine-tuned VAE to reveal the superiority of our proposed flow-based paradigm.

**Direct Mask Prediction (Worst Performance).** A direct, single-step mapping from the video and text latents to the mask latent represents the classic discriminative paradigm. We argue this approach is fundamentally ill-posed due to what we term "information collapse." The mapping from a high-entropy, complex video manifold to a low-entropy, sparse mask manifold is a drastic information contraction. Forcing a neural network to learn this in a single, abrupt step leads to collapsion of the rich visual context into a coarse approximation rather than performing a precise, guided refinement. The model is not learning a transformation, but rather a brittle pattern recognition function.

**Noise-to-Mask Flow (Suboptimal).** This paradigm mirrors standard text-to-video generation, starting from Gaussian noise $\mathbf{z}_1 \sim \mathcal{N}(0, I)$ and conditioning on the video context. This approach demotes the video from the primary source of information to a secondary condition. Weaken the guidance of the video in the process would possibly force the entire, high-dimensional video context to be injected via a simple concatenation at each step, creating a severe information bottleneck. The model is tasked with generating the mask's complex spatio-temporal structure from scratch based on this limited conditional signal, rather than progressively refining the rich, structured information already present in the video itself.

**One-Step Velocity Prediction (Better, but Limited).** This paradigm makes the model predict the full velocity vector $\mathbf{v} = \mathbf{z}_1 - \mathbf{z}_0$ in a single inference step. It significantly outperforms the previous two baselines, confirming our hypothesis that learning a residual velocity is a more stable and effective objective than predicting an absolute state. However, its performance is still fundamentally capped. We assume that it is limited by the need to compute the entire, often large-magnitude deformation in a single forward pass, lacking the capacity for the gradual, iterative refinements that a multi-step process allows.

This analysis solidifies our central thesis: a multi-step, video-to-mask flow is the most effective paradigm for RVOS, but only when augmented with our proposed start-point focused adaptations to bridge the critical gap between its generative origins and the discriminative demands of the task.

## 4 EXPERIMENTS

### 4.1 BENCHMARK AND METRICS

We evaluate our framework on three standard RVOS benchmarks. MeViS (Ding et al., 2023) is a challenging, motion-centric benchmark featuring 2,006 long videos and over 28,000 fine-grained annotations that emphasize complex dynamics. Ref-YouTube-VOS (Wu et al., 2022) is the large-scale benchmark, comprising 3,978 videos that test for generalizability across a wide diversity of objects and scenes. Ref-DAVIS17 (Khoreva et al., 2018) is a high-quality, densely annotated dataset of 90 videos, serving as a key benchmark for segmentation precision and temporal consistency.

Following standard protocols, we report region similarity ($\mathcal{J}$, Jaccard Index), contour accuracy ($\mathcal{F}$, F-measure), and their average ($\mathcal{J}\&\mathcal{F}$) as our primary evaluation metrics.

### 4.2 IMPLEMENTATION DETAILS

Our framework is built upon the publicly available Wan 2.1 text-to-video model, which features a 1.3B parameter Diffusion Transformer (DiT) (Wan et al., 2025). Throughout all training stages, we keep the pretrained text encoder and the VAE encoder entirely frozen. Our training focuses exclusively on fine-tuning the DiT block to learn the conditional flow. Crucially, the VAE decoder is specifically adapted for the segmentation task by being fine-tuned separately on the MeViS training set, allowing it to specialize in reconstructing high-quality masks from the latent space. Our training protocol varies by dataset to align with the same evaluation protocols of compared methods. For experiments on Ref-YouTube-VOS, we follow a two-stage training strategy. The model is first pre-trained on a combination of static image datasets (RefCOCO/+/g) (Yu et al., 2016; Kazemzadeh et al., 2014) to learn foundational visual-linguistic grounding. Subsequently, this pre-trained model

Table 1: Comparison of our one-stage FlowRVS with other previous 'locate-then-segment' methods on MeViS, Ref-YouTube-VOS and Ref-DAVIS datasets. We further include methods based on large VLMs for comparison. **Bold** and underline indicate the two top results.

| Method | MeViS | | | Ref-YouTube-VOS | | | Ref-DAVIS17 | | |
|---|---|---|---|---|---|---|---|---|---|
| | $\mathcal{J}\&\mathcal{F}$ | $\mathcal{J}$ | $\mathcal{F}$ | $\mathcal{J}\&\mathcal{F}$ | $\mathcal{J}$ | $\mathcal{F}$ | $\mathcal{J}\&\mathcal{F}$ | $\mathcal{J}$ | $\mathcal{F}$ |
| *Locate-then-segment* | | | | | | | | | |
| MTTR [CVPR'22] | 30.0 | 28.8 | 31.2 | 55.3 | 54.0 | 56.6 | - | - | - |
| ReferFormer [CVPR'22] | 31.0 | 29.8 | 32.2 | 62.9 | 61.3 | 64.6 | 61.1 | 58.1 | 64.1 |
| SOC [NIPS'23] | - | - | - | 66.0 | 64.1 | 67.9 | 64.2 | 61.0 | 67.4 |
| OnlineRefer [ICCV'23] | 32.3 | 31.5 | 33.1 | 63.5 | 61.6 | 65.5 | 64.8 | 61.6 | 67.7 |
| LISA [CVPR'24] | 37.2 | 35.1 | 39.4 | 53.9 | 53.4 | 54.3 | 64.8 | 62.2 | 67.3 |
| DsHmp [CVPR'24] | 46.4 | 43.0 | 49.8 | 67.1 | 65.0 | 69.1 | 64.9 | 61.7 | 68.1 |
| VideoLISA [NIPS'24] | 42.3 | 39.4 | 45.2 | 61.7 | 60.2 | 63.3 | 67.7 | 63.8 | 71.5 |
| VD-IT [ECCV'24] | - | - | - | 64.8 | 63.1 | 66.6 | 63.0 | 59.9 | 66.1 |
| VISA [ECCV'24] | 43.5 | 40.7 | 46.3 | 61.5 | 59.8 | 63.2 | 69.4 | 66.3 | 72.5 |
| SSA [CVPR'25] | 48.9 | 44.3 | 53.4 | 64.3 | 62.2 | 66.4 | 67.3 | 64.0 | 70.7 |
| SAMWISE [CVPR'25] | 49.5 | 46.6 | 52.4 | 69.2 | **67.8** | 70.6 | 70.6 | 67.4 | 74.5 |
| ReferDINO [ICCV25] | 49.3 | 44.7 | 53.9 | 69.3 | 67.0 | 71.5 | 68.9 | 65.1 | 72.9 |
| *One-stage generation based* | | | | | | | | | |
| **FlowRVS (ours)** | **51.1** | **47.6** | **54.6** | **69.6** | 67.1 | **72.1** | **73.3** | **68.4** | **78.2** |

is fine-tuned on the Ref-YouTube-VOS training set. The final weights from this stage are then used for zero-shot evaluation on the Ref-DAVIS17 benchmark without any further fine-tuning to prove FlowRVS's generalization ability. For the more challenging MeViS dataset, which emphasizes complex motion understanding, we train our model directly on its training set from scratch, without leveraging any static image pre-training. More hyperparameters settings can be found in Appendix A.

## 4.3 MAIN RESULTS

We compare our proposed FlowRVS against a wide range of baselines in Table 1. Our results show that FlowRVS significantly and consistently outperforms previous approaches. We highlight some key features as follows:

**Dominance on Complex Motion-Centric Benchmarks.** The most significant advantage of our framework is demonstrated on MeViS, the most challenging benchmark designed to test a nuanced understanding of motion-centric language. FlowRVS achieves a $\mathcal{J}\&\mathcal{F}$ score of 50.7, establishing a new SOTA and surpassing the previous best method, SAMWISE, by a substantial 1.2 point margin. This result is particularly noteworthy as MeViS features long videos with complex object interactions and appearance changes—scenarios where the limitations of multi-stage, cascaded pipelines are most exposed. The superior performance of FlowRVS directly validates our core thesis: a holistic, end-to-end flow that models the entire video-to-mask transformation is fundamentally better suited to capture and reason about complex spatio-temporal dynamics.

**Superiority over 'Locate-then-Segment' Paradigms**. Our performance gains are particularly meaningful when compared directly against methods that epitomize the 'locate-then-segment' paradigm, such as VISA (VLM-based) and ReferDINO (grounding-model-based). On MeViS, FlowRVS outperforms VISA-13B by a remarkable 7.0 points and ReferDINO (best results announced in the paper) by 1.4 points. This underscores the advantage of our one-stage approach. By avoiding the irreversible information loss inherent in collapsing semantics into an intermediate geometric or feature prompt, our continuous flow process maintains a high-fidelity, text-guided transformation from start to finish, leading to more accurate and robust segmentation. The fundamental advantages of our one-stage flow paradigm are further illustrated in our qualitative comparisons (Figure 4). For the query "The white rabbit which is jumping," ReferDINO provides a coarse, static grounding of the rabbit but misses the jumping action's details, whereas FlowRVS delivers a precise, dynamic segmentation. More critically, for the temporal query "The first tiger...", VD-IT's

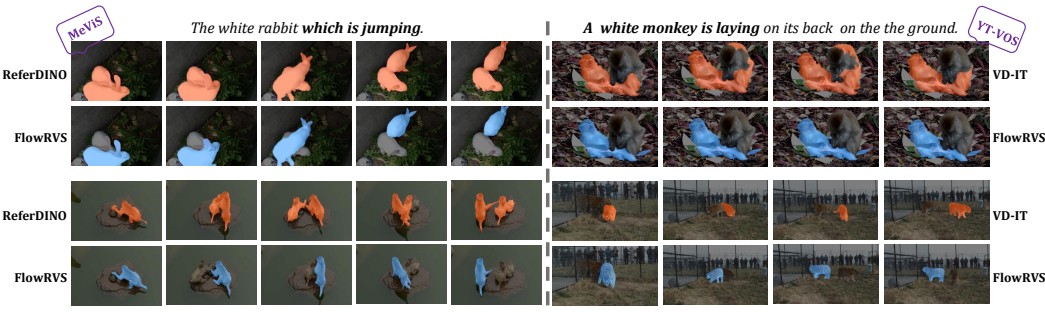

Figure 4: Qualitative comparison on challenging temporal and linguistic reasoning. Prior paradigms struggle: VD-IT produces temporally unstable masks due to its frame-wise decoder, while Refer-DINO fails to interpret long-range descriptions. Our method, FlowRVS, demonstrates superior temporal coherence and language grounding by leveraging an end-to-end generative process.

Table 2: Ablations of FlowRVS on the MeVIS $valid_u$ set. BBS: Boundary-Biased Sampling (probability $p$). DVI: Direct Video Injection. SPA: Start Point Augmentation. WI: Weight Init from Wan.

| ID | Method Configuration | BBS ($p$) | SPA | DVI | WI | $\mathcal{J}\&\mathcal{F}$ | $\mathcal{J}$ | $\mathcal{F}$ |
|---|---|---|---|---|---|---|---|---|
| *Alternative Paradigms* | | | | | | | | |
| (a) | MultiStep Noise-to-Mask Flow | – | – | ✓ | ✓ | 32.3 | 29.6 | 35.0 |
| (b) | Onestep Mask Prediction | – | – | – | ✓ | 38.9 | 36.2 | 41.5 |
| (c) | Onestep Velocity Prediction | – | – | – | ✓ | 50.8 | 47.1 | 54.4 |
| *Our MultiStep Video-to-Mask Flow* | | | | | | | | |
| (c) | Base Flow | 0.0 | – | – | ✓ | 47.9 | 42.9 | 52.9 |
| (d) | + BBS | 0.25 | – | – | ✓ | 55.2 | 50.7 | 59.6 |
| (e) | + BBS | 0.50 | – | – | ✓ | 57.9 | 53.8 | 62.1 |
| (f) | + BBS | 0.75 | – | – | ✓ | 56.5 | 52.5 | 60.4 |
| (g) | + SPA | 0.50 | ✓ | – | ✓ | 58.6 | 54.2 | 63.0 |
| (h) | + DVI (**ours default**) | **0.50** | ✓ | ✓ | ✓ | **60.6** | **55.9** | **65.2** |
| (i) | - WI | 0.50 | ✓ | ✓ | ✗ | 21.1 | 20.3 | 21.9 |

decoupled decoder fails to resolve the ambiguity and tracks the wrong target. In contrast, FlowRVS correctly identifies and tracks the first tiger throughout, demonstrating superior global reasoning.

**Exceptional Zero-Shot Generalization on Ref-DAVIS17.** The generalization capability of our model is best illustrated by its zero-shot performance on Ref-DAVIS17. Without any fine-tuning on the DAVIS dataset, the model trained on Ref-YouTube-VOS achieves a $\mathcal{J}\&\mathcal{F}$ score of 73.3. This result is not only state-of-the-art but is also significantly higher than many previous methods that were explicitly trained or fine-tuned on similar high-quality datasets. This strong zero-shot transferability suggests that our flow-based paradigm, by learning a more fundamental and continuous mapping between video and its corresponding mask guided by language, develops a more generalizable understanding of spatio-temporal correspondence that is less prone to dataset-specific biases.

### 4.4 ABLATION STUDIES

We conduct our ablation studies on the challenging MeViS dataset, as its complex, motion-centric scenarios provide a rigorous testbed for our design choices. To ensure a consistent and fair comparison, all results are reported on the $valid_u$ set, following the protocol in prior work (Ding et al., 2023). The results are summarized in Table 2.

**Analysis of Alternative Paradigms.** Our investigation begins by establishing the limitations of alternative paradigms (rows a-c). The Noise-to-Mask Flow (a), which mirrors standard generative practices, performs poorly (32.3 $\mathcal{J}\&\mathcal{F}$). This confirms our hypothesis that demoting the video to

Table 3: Analysis of VAE adaptation strategies. We measure both the mask reconstruction quality (Recon.) and the resulting performance (Perf.) on the MeViS $valid_u$ set with a fixed flow model. VAE is tuned on MeViS training set.

| VAE Adaptation Strategy | Recon. $\mathcal{J}\&\mathcal{F}$ | Perf. $\mathcal{J}\&\mathcal{F}$ |
|---|---|---|
| Frozen VAE | 98.8 | 60.0 |
| Finetuning Decoder (**ours default**) | **99.1** | **60.9** |

a secondary condition (concatenate with noise) creates a severe information bottleneck, forcing the model to generate the mask from scratch. The Onestep Mask Prediction model (b) also struggles (36.2 $\mathcal{J}\&\mathcal{F}$), validating that a single, abrupt mapping is insufficient to bridge the vast representational chasm between video and mask. Notably, shifting the objective from state prediction to Onestep Velocity Prediction (c) yields a substantial +14.6 $\mathcal{J}\&\mathcal{F}$ gain. This key result proves that learning a residual (velocity) is a fundamentally more stable and effective task, providing strong initial validation for our flow-based reformulation.

**Effectiveness of Start-Point Focused Adaptations.** Having confirmed the video-to-mask flow as the most promising direction, we dissect our proposed adaptations (rows c-h). The Base Flow model (c), trained with naive uniform sampling, performs poorly at 47.9 $\mathcal{J}\&\mathcal{F}$, even worse than the one-step velocity predictor. This confirms that a multi-step process is not inherently superior; it must be correctly stabilized. The introduction of **Boundary-Biased Sampling (BBS)** provides the definitive solution. As shown in rows (d)-(f), forcing the model to focus on the start of the trajectory by oversampling $t = 0$ almost single-handedly unlocks the potential of the multi-step flow. Even a moderate bias of $p = 0.25$ (d) brings a massive +7.3 point improvement. The performance peaks at $p = 0.5$ (e), yielding a total gain of +10.0 $\mathcal{J}\&\mathcal{F}$ over the baseline. While a more extreme bias of $p = 0.75$ (f) leads to a slight performance drop, the score of 56.5 remains substantially higher than the baseline, demonstrating that the strategy is robust and effective across a reasonable range of hyperparameters. This confirms that mastering the initial, text-guided velocity is the most critical factor for success. Finally, Direct Video Injection (DVI) (h) provides a persistent context anchor throughout the trajectory, preventing drift and adding a significant +2.0 $\mathcal{J}\&\mathcal{F}$.

**Effectiveness of the T2V Pretrain Model.** Finally, we validate the central premise of our work: leveraging the power of large-scale T2V models. As shown in row (i), training our model from scratch without the pretrained weights (**-WI**) results in a complete performance collapse to 21.1 $\mathcal{J}\&\mathcal{F}$. This underscores that our contributions are not generic training method, but are specifically designed to effectively harness and adapt the powerful priors learned by generative foundation models for this challenging discriminative task.

**VAE Adaptation and frozen VAE.** A crucial step in our method is adapting the pretrained VAE to accurately transform between the latent space and the pixel space of binary masks. As shown in Table 3, we evaluate the effectiveness of the decoder adaptation. Consistent with observations in concurrent work REM (Bagchi et al., 2025), we find that the frozen VAE decoder can already support competitive segmentation performance (60.0 $\mathcal{J}\&\mathcal{F}$). However, fully finetuning the decoder further bridges the domain gap between continuous video latents and binary masks. This adaptation yields superior reconstruction quality and provides a solid performance gain (+0.9 $\mathcal{J}\&\mathcal{F}$) over the frozen baseline.

## 5 CONCLUSION AND FUTURE WORK

In this work, we introduce FlowRVS, a framework that moves beyond using T2V models as mere feature extractors and instead reformulates RVOS as a continuous, text-conditioned flow from video to mask. Our core contribution is demonstrating that this paradigm shift, when combined with our proposed start-point focused adaptations (BBS, SPA, DVI), successfully aligns the generative strengths of T2V models with the discriminative demands of the task, leading to state-of-the-art performance. Our findings validate that the key to unlocking these models lies in principled adaptation, proving that by fortifying the flow's structured starting point, the philosophical gap between generative processes and discriminative objectives can be effectively bridged.

Looking forward, we believe the paradigm of modeling understanding tasks as conditional deformation processes holds significant potential beyond RVOS. And our insight in stabilizing discriminative, start-point-critical flows provide a crucial blueprint for the future. As even bigger and more powerful foundation models emerge, these techniques will be essential for harnessing their full potential and applying their remarkable capabilities to the vast amount of video understanding tasks.

## 6 ACKNOWLEDGMENT

This work was supported by Zhejiang Province Natural Science Foundation of China under Grant No. LQN25F030008 and the National Natural Science Foundation of China under Grant No. 62403429.

## ETHICS STATEMENT

This manuscript does not cover any topics or experiments related to ethics.

## REPRODUCIBILITY STATEMENT

This manuscript provides detailed implement details, hyperparameter settings for reproduction. In addition, the code will be open-sourced later at depended on acceptance

## USE OF LLMS

We leverage large language models (LLMs) to draft and polish our manuscripts, only for ensuring each sentence is clearer and the narrative flows coherently.

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

APPENDIX

# A    HYPERPARAMETERS SETTINGS

All our models are trained using the AdamW optimizer. We detail the key hyperparameters for our main experiments on Ref-YouTube-VOS and MeViS in Table 4.

Table 4: Key hyperparameters for the training of FlowRVS. We detail the settings for the 2D pre-training, the main DiT fine-tuning on video datasets, and the separate VAE decoder adaptation.

| Hyperparameter | 2D Pre-training (RefCOCO+/g) | Video DiT Fine-tuning (Ref-YT-VOS & MeViS) | VAE Decoder Fine-tuning (on MeViS) |
|---|---|---|---|
| *Optimizer Configuration* | | | |
| Optimizer | AdamW | AdamW | AdamW |
| Peak Learning Rate | 7e-5 | 6e-5 | 8e-5 |
| LR Schedule | None | None | None |
| Warmup Steps | 0 | 0  0 | 0 |
| Weight Decay | 5e-4 | 5e-4 | 5e-4 |
| Adam $\beta_1, \beta_2$ | (0.9, 0.999) | (0.9, 0.999) | (0.9, 0.999) |
| *Training Schedule & Loss* | | | |
| Total Training Epochs | 6 | 7 / 6 | 1 |
| Global Batch Size | 8 | 8 | 4 |
| Per-GPU Batch Size | 1 | 1 | 1 |
| Gradient Accumulation Steps | 1 | 1 | 1 |
| Mixed Precision | bfloat16 | bfloat16 | bfloat16 |
| Loss Function | L2 (MSE Loss) | L2 (MSE Loss) | Combined Focal + Dice Loss ($\alpha = 0.25, \gamma = 2.0$) |

# B    MORE QUALITATIVE RESULTS

In this section, we provide additional qualitative results of FlowRVS on challenging video-text pairs. These examples further demonstrate that our holistic, flow-based approach successfully handles complex language and dynamic scenes, particularly in scenarios involving significant occlusion and nuanced textual descriptions.

# C    COMPREHENSIVE RESULTS

While our method demonstrates strong performance, it is not without limitations. We present two typical failure modes on Figure 8. For complex relational phrases requiring fine-grained interaction understanding, such as "swings its tail and strikes the head" on the top row, the model correctly identifies the primary subject (the horse) but fails to isolate the specific horse performing the action. This suggests a limitation in comprehending intricate multi-part actions. In scenarios with multiple similar objects and temporal ordering cues ("the second bird") on the bottom row, the model struggles to accurately resolve the ambiguity. It incorrectly segments the first bird that moves, indicating that while our temporal modeling is strong, it can be confounded by challenging counting and ordering logic within dense scenes.

To address the question of generalization to unseen actions and challenging description, we tested our model on phrases not present in the training data, such as "the dog somersaulting." As shown on Figure 9, FlowRVS successfully identifies and segments the dog throughout its complex, non-rigid motion. This demonstrates that our method does not merely memorize action-object pairings from the training set. Instead, by leveraging the rich spatio-temporal and semantic priors from the pretrained T2V model, it develops a more fundamental understanding based on open-set vocabulary queries that allows it to generalize to novel and dynamic actions.

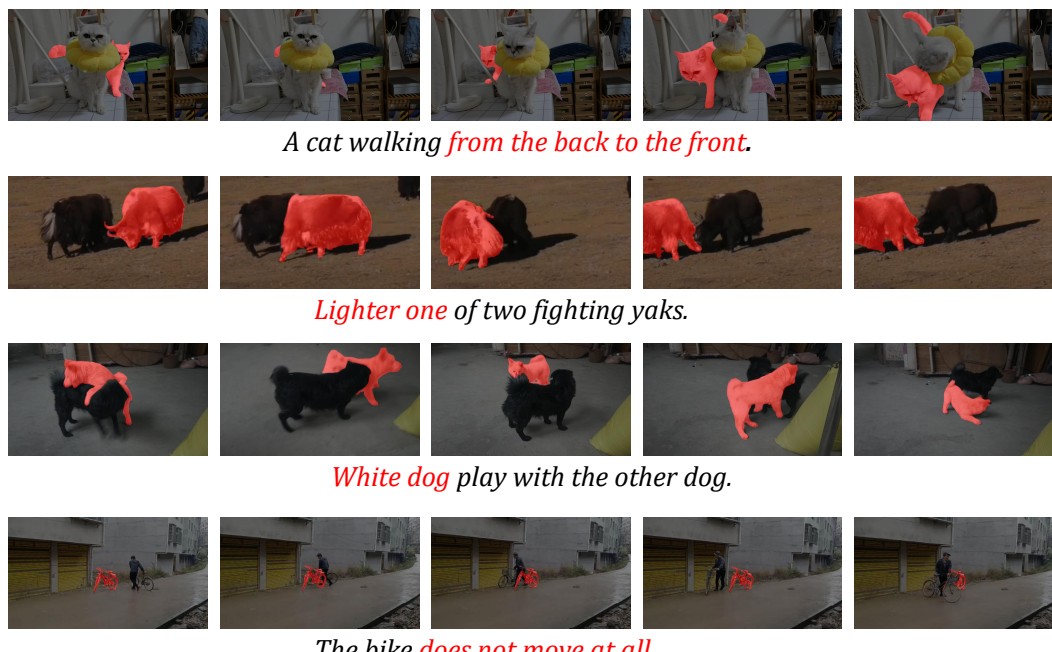

*A cat walking from the back to the front.*

*Lighter one of two fighting yaks.*

*White dog play with the other dog.*

*The bike does not move at all.*

Figure 5: visualization example of FlowRVS results on MeViS-Valid-u.

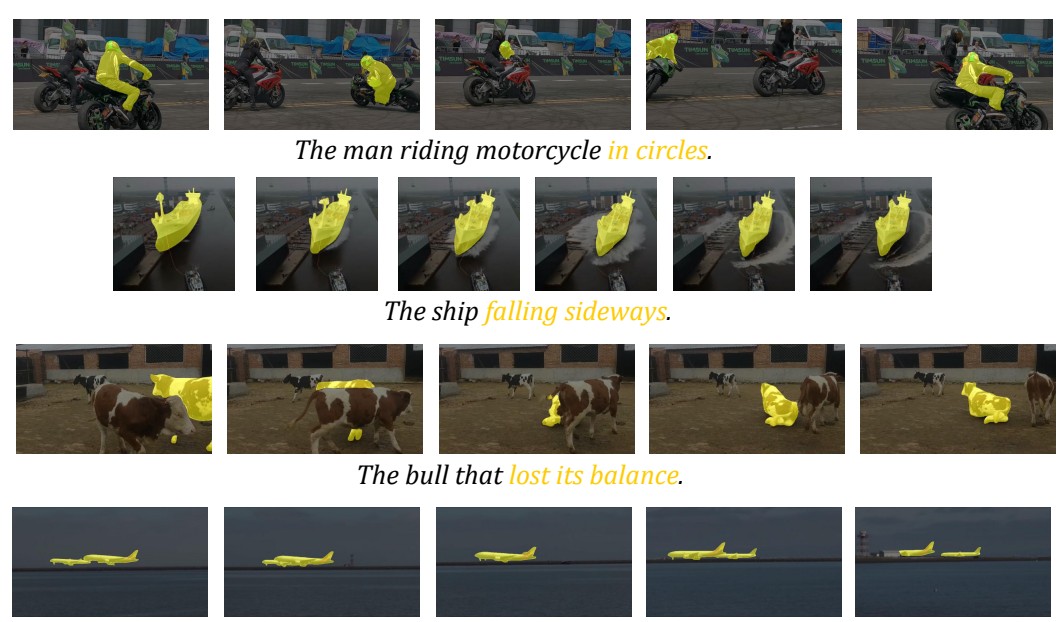

*The man riding motorcycle in circles.*

*The ship falling sideways.*

*The bull that lost its balance.*

*The two aircrafts navigating above the sea surface and preparing to land.*

Figure 6: visualization example of FlowRVS results on MeViS-Valid.

# D    DETAILS ABOUT PROPOSED IMPROVEMENTS

## D.1    BOUNDARY-BIASED SAMPLING (BBS)

BBS is a training strategy to emphasize the crucial initial step of the flow ($t = 0$). Formally, we sample the timestep $t$ from a mixed probability distribution, whose probability density function $f(t)$ is defined as: $f(t) = p \cdot \delta(t) + (1-p) \cdot \mathcal{U}(t|0, 1)$ where $\delta(t)$ is the Dirac delta function representing

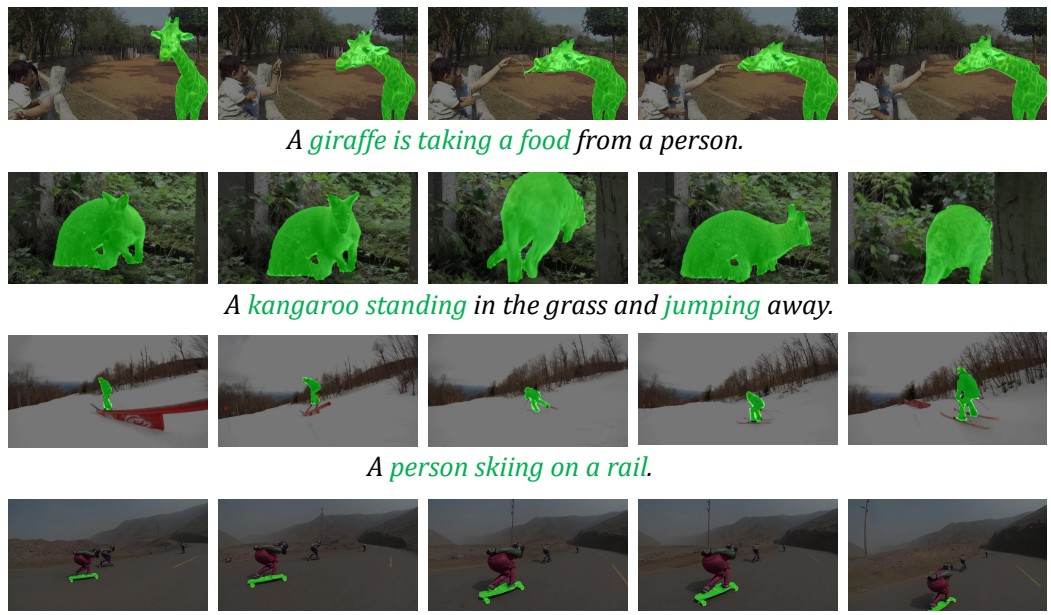

*A giraffe is taking a food from a person.*

*A kangaroo standing in the grass and jumping away.*

*A person skiing on a rail.*

*A skateboard is being rode by a person wearing red pants up the hill.*

Figure 7: visualization example of FlowRVS results on Ref-YouTube-VOS-Valid.

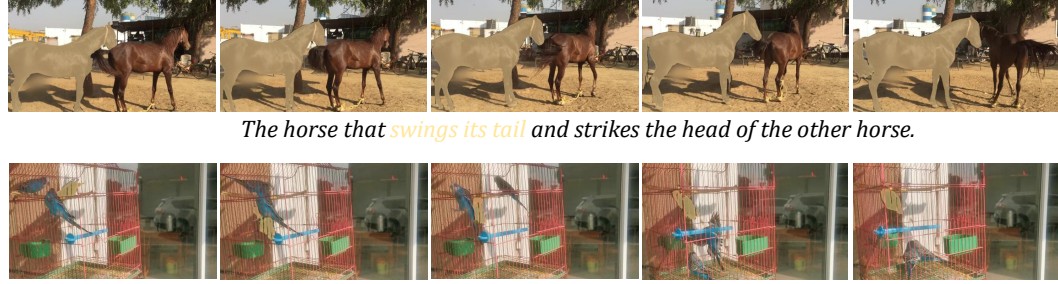

*The horse that swings its tail and strikes the head of the other horse.*

*The second bird to reach the bottom of the cage.*

Figure 8: failure cases of FlowRVS results.

a point mass at $t = 0$, $\mathcal{U}(t|0, 1)$ is the uniform distribution on the interval $[0, 1]$, and $p$ is the bias probability. In practice, this means we sample $t = 0$ with probability (p=0.5).

### D.2 START-POINT AUGMENTATION (SPA)

SPA is a crucial regularizer to improve the robustness of our convergent (video-to-mask) flow. For a given video $V$, the VAE encoder $E_\phi$ predicts a posterior distribution $q(z|V) = \mathcal{N}(z|\boldsymbol{\mu}_V, \boldsymbol{\sigma}_V^2)$, where $\boldsymbol{\mu}_V$ and $\boldsymbol{\sigma}_V^2$ are directly inherited from the VAE. Instead of using the deterministic mean $\boldsymbol{\mu}_V$, SPA samples the starting point $\boldsymbol{z}_0'$ from this posterior: $\boldsymbol{z}_0' \sim \mathcal{N}(\boldsymbol{z}|\boldsymbol{\mu}_V, \boldsymbol{\sigma}_V^2)$. This sampled latent $\boldsymbol{z}_0'$ is then normalized before being used the ODE solver. By augmenting the training data with samples from the local neighborhood of each video's true latent representation, SPA forces the model to learn a smoother and more generalizable velocity field.

### D.3 DIRECT VIDEO INJECTION (DVI)

DVI provides the model with a persistent anchor to the original video content throughout the ODE trajectory. It is implemented by concatenating the current state $\boldsymbol{z}_t$ with the initial video latent $\boldsymbol{z}_0$. This changes the input tensor's shape from $[B, C, T, H, W]$ to $[B, 2 * C, T, H, W]$. This is handled

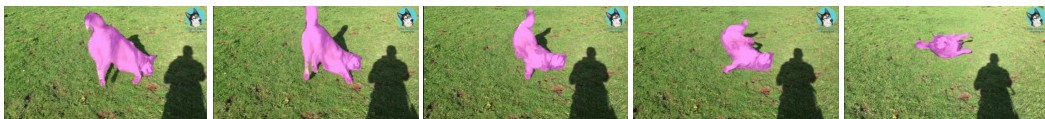

*The dog somersaulting.*

Figure 9: novel action phrases out of MeViS training.

by modifying the first convolutional layer of the DiT to accept $2 * C$ input channels, while all subsequent layers remain unchanged.

