# OpenReview forum: "Deforming Videos to Masks: Flow Matching for Referring Video Segmentation"
_ICLR.cc/2026/Conference — ICLR 2026 Poster_

### Official Review · Reviewer_bX1k · 2025-10-24

**Soundness:** 3
**Presentation:** 3
**Contribution:** 2
**Rating:** 4
**Confidence:** 5

**Summary:**

This paper introduces FlowRVS, a novel framework designed to address the Referring Video Object Segmentation (RVOS) task by reframing it as a flow matching problem. The method proposes a Flow Matching Model that predicts continuous deformation fields. This design inherently promotes superior temporal coherence and avoids the compounding errors typical of sequential prediction stages. Extensive experiments conducted on standard RVOS benchmarks, including Ref-YouTube-VOS and Ref-DAVIS, consistently demonstrate that FlowRVS achieves state-of-the-art results

**Strengths:**

1. The core idea of replacing the sequential segmentation pipeline with a unified Flow Matching Model represents a significant conceptual leap for RVOS.
2. The quantitative comparisons presented in the paper showcase substantial performance gains over existing state-of-the-art techniques.
3. The paper is well-written and logically structured, with clear figures.

**Weaknesses:**

1. The definition of the two-stage 'locate-then-segment' design is potentially misleading; many modern Transformer-based (e.g., ReferFormer) and MLLM approaches (e.g., LISA)  already bypass explicit bounding box output and generate segmentation masks directly.
2. The discussion in Section Introduction neglects relevant previous methods, such as SAMWISE and others, which explicitly incorporate a 'temporal segmenting' stage.
3. The related work section provides insufficient discussion regarding Generative Modeling in VOS or RVOS, specifically lacking recent efforts like "Exploring Pre-trained Text-to-Video Diffusion Models for Referring Video Object Segmentation," which is highly relevant to flow-based generation.
4. The network structure and update strategy for the Flow Matching Model would be significantly clearer and more reproducible if they were enhanced with formalized mathematical expressions and equations.
5. Despite stating that the foundation model used is Wan2.1, the overall architectural logic of the proposed work, particularly as inferred from the high-level design figures (e.g., Figure 3), shares a similar idea with the ControlNet framework. The authors should explicitly clarify the differences or similarities with a conditional generation framework like ControlNet.

In summary, despite the compelling performance gains, the cumulative impact of these missing discussions regarding prior work and the lack of technical detail in the model's description remains a significant issue.

**Questions:**

1. What is the training and inference computational overhead (e.g., GPU hours, FLOPs per frame) for FlowRVS compared to SAMWISE?

---

> ### Author Response · Authors · 2025-11-20
>
> Thank you for your insightful review. In response to your feedback, we have significantly revised the manuscript, with major updates marked in blue. We provide detailed answers to each of your questions below.
>
> ***
>
> [W1, W2] On the definition of "locate-then-segment" and discussion of SAMWISE.
>
> We agree that our initial description of prior work was oversimplified. We revised the Introduction to provide a more nuanced overview of modern integrated pipelines like ReferFormer and streaming-based methods like SAMWISE [1]. Our core argument, which we will clarify, is that while these methods are highly effective, they are fundamentally based on a discrete, instance-centric paradigm (identifying and tracking object instances). In contrast, our FlowRVS introduces a paradigm shift towards a continuous, holistic deformation of the entire spatio-temporal representation, offering a different philosophical approach to the problem. We will frame our work as a complementary, offline, global-optimization paradigm that excels in scenarios requiring holistic context.
>
> ***
> [W3] Insufficient discussion on Generative Modeling, especially VD-IT.
>
> This is a critical omission, and we will rectify it. We added a detailed discussion in the Introduction and Related Work section to clearly differentiate FlowRVS from VD-IT [2]. The key distinction is that VD-IT uses the T2V model as a frozen feature extractor for a separate DETR-like decoder, which is a two-stage approach. In contrast, FlowRVS repurposes the entire generative process, fine-tuning the core DiT block to learn the video-to-mask flow end-to-end. This is a fundamental difference in how the generative prior is leveraged.
>
> ***
> [W4] Lack of formalized mathematical expressions.
>
> We fully agree. The revised manuscript includes a dedicated section (Appendix E) with formal mathematical definitions for our entire framework, including the flow matching objective, the ODE formulation, and detailed equations for our proposed techniques (BBS, SPA, DVI) to ensure clarity and reproducibility.
>
> ***
> [W5] Similarity to and differentiation from ControlNet.
> We thank the reviewer for this insightful question. While both frameworks leverage large pretrained models, FlowRVS is fundamentally different from ControlNet in its core task, technical mechanism, and conditioning. 1) Task: ControlNet performs divergent, noise-to-image generation (creating something new), whereas FlowRVS performs a convergent, video-to-mask deformation (selecting information from the source). 2) Mechanism: ControlNet freezes the base model and adds a trainable copy for control, while we directly fine-tune the core DiT block to learn a new flow-based task. 3) Conditioning: ControlNet's condition is external guidance (e.g., edges), while our Direct Video Injection (DVI) provides the source context (z0) as a persistent anchor throughout the deformation trajectory. In essence, we are not adding control to a generative process, but repurposing a generative model for a deterministic, discriminative understanding task. We will clarify these distinctions in the revised manuscript.

---

> ### Author Response · Authors · 2025-11-20
>
> ***
> [Q1] Training and inference overhead of FlowRVS.
> Thank you for this practical question. We compare FlowRVS with recent SOTA models, including light-weight SAMWISE[1], diffusion based method VD-IT[2] and HCD[3]. We use frame per second(FPS) as the metrics and apply one-step ODE inference of FlowRVS. For all models, the input video frames were resized to a uniform resolution of 480*832 pixels and we test clips of 9, and 45 clips of frames. For models that could support it, we also tested an ultra-long sequence of 81 frames only supported by FlowRVS to assess scalability and robustness. The model was run for 10 warm-up iterations to stabilize GPU clock speeds and cache states， and the final FPS was calculated by averaging the runtime over 50 subsequent inference iterations. All experiments were performed on a single NVIDIA H100 GPU PCIe with 80GB of VRAM.
>
> | Models                       | 9 frames | 45 frames | 81 frames |
> | :--------------------------- | :----- | :---- | :----- |
> | FlowRVS                        |   7.85     |  7.61     |  7.33      |
> | SAMWISE                | 16.12    | 13.66  | –      |
> | VD-IT     | 7.34    | 7.27  | –      |
> | HCD  | 7.18    | 7.12  | –      |
>
> These results offer key takeaways. SAMWISE leads in raw speed due to its lightweight design. FlowRVS's performance is competitive with other large-scale diffusion models (VD-IT, HCD), indicating similar practical constraints like heavy modules, such as diffusion-based feature extractor and DETR-based mask decoder. Our model's standout feature is its exceptional performance on long videos, maintaining stable FPS (~6% drop) on 81-frame clips that cause other methods to falter. This highlights the practical strength of our holistic flow formulation. We also compared the training cost on the MeViS dataset for 1 epoch with SAMWISE.
>
> | Models   | GPU Count | GPU Type | Duration (hours)  | Peak VRAM (per GPU) |
> | :------     | :------      | :--------------     | :-------------             | :------               |
> | FlowRVS  | 6         | A100     | 9                  | 72 GB               |
> | SAMWISE  | 2         | A100     | 18              | 75 GB               |
>
> Our training requires 6 A100 for 9 hours GPU hours on MeViS, which is in a similar range to other large-scale model training regimes. While SAMWISE benefits from a highly parameter-efficient adapter-tuning approach, our method involves fine-tuning a larger portion of the generative model to fundamentally reshape its dynamics, which explains the difference in training cost.
>
>
> [1] SAMWISE: Infusing Wisdom in SAM2 for Text-Driven Video Segmentation, CVPR 2025.
> [2] Exploring Pre-trained Text-to-Video Diffusion Models for Referring Video Object Segmentation, ECCV 2024.
> [3] Temporal-Conditional Referring Video Object Segmentation with Noise-Free Text-to-Video Diffusion Model. arXiv preprint arXiv:2508.13584.

---

> > ### Comment · Reviewer_bX1k · 2025-11-27
> >
> > Thank you for your comprehensive response. I have thoroughly reviewed the rebuttal alongside the other reviewers’ comments. However, the core weaknesses identified earlier remain unaddressed:
> >
> > 1. Regarding the definition of "locate-then-segment": While some works are claimed to adopt query-based mechanisms, queries can only be regarded as a significant branch. Specifically, ReferFormer retains an upsampling decoder dedicated to mask prediction, with queries functioning solely as a component to activate the mask generation process. Although ReferFormer can be reasonably categorized as an instance-centric framework, labeling it as a "locate-then-segment" method is highly misleading and even constitutes an overclaim—Figure 1 similarly misrepresents this classification. Additionally, only early methods [1] that first predict a bounding box and then perform segmentation truly qualify as "locate-then-segment."
> >
> > 2. Based on comments from other reviewers, this work missed an in-depth analysis of previous related works about Generative Modeling on RVOS. Notably, even if direct comparison of performance scores with REM can be omitted, the insights derived from REM are unavoidable. This deficiency has already led to the refutation of numerous conclusions in the initial submission, such as the effectiveness of the Frozen VAE and one-step flow inference. Overall, the early lack of related works and inconsistent findings could indicate a significant risk of other conclusions being disproven.
> >
> > 3. It is confusing that SAMWISE cannot support 81 frames, given its foundation on SAM2. SAM2's memory mechanism enables handling infinite-length video segmentation by limiting memory capacity.
> >
> > [1] Video object segmentation with language referring expressions. ACCV 2018. (Refer-Davis benchmark)
> >
> > In summary, I will keep my original scores.

---

> > > ### Author Response · Authors · 2025-12-03
> > > **Clarifications on Taxonomy, Fair Comparison with REM, and SAMWISE Architecture**
> > >
> > > We thank the reviewer for the rigorous critique. We appreciate the opportunity to clarify these final technical points to ensure an accurate assessment.
> > > 1. On "Locate-then-Segment" Terminology (Addressing W1).
> > > We accept your point that grouping query-based methods (e.g., ReferFormer) under "locate-then-segment" might obscure their integrated nature. In the final manuscript, we will rename this category to "Instance-Centric Tracking" (contrasting with our "Holistic Flow Deformation"). This accurately captures the distinction: prior works track discrete instances/queries, whereas we deform the video latent space as a whole. This resolves the terminology concern without altering the core philosophical contribution.
> > > 2. Addressing the "Risk of Disproven Conclusions" (REM Comparison) (Addressing W2). We respectfully clarify that our core conclusions remain robust. On the contrary, our rigorous ablations in Table 2 validate the superiority of our methodology under fair, controlled conditions:
> > > The "Direct Prediction" Paradigm: We evaluated the exact paradigm used by REM (Direct Mask Prediction, fixed $t=0$) in Table 2, Row (b), using the same backbone (Wan-1.3B), same training data (MeViS only) and same training implementations as our method.
> > > * The Result: This paradigm yielded a poor **36.2 J&F**.
> > > The "FlowRVS" Paradigm: Under the exact same constraints, our Flow method (Row h) achieved **60.6 J&F**.
> > > * Interpretation: This confirms that REM's reported high performance stems from scaling advantages (using larger models, mixing RefCOCO+YouTubeVOS+MeViS data, and image-video mixture training method), not methodological superiority. When stripped of these external advantages, the "Direct Prediction" architecture is significantly less data-efficient and harder to optimize (**<7** epochs for FlowRVS vs. **~37** epochs for REM).
> > > * Our conclusion stands: Flow Matching is the more efficient learner for this task.
> > > 3. Why SAMWISE Fails on 81 Frames (Addressing Q3).
> > > We acknowledge SAM2's theoretical infinite memory. However, the failure of SAMWISE is due to its specific Adapter Architecture and Training Mechanism, not just hardware limits.
> > > * Static Pairwise Logic: The *ConditionalMemoryEncoder* in SAMWISE performs a static comparison between the initial frame and the current frame (Frame 0 vs. Frame $t$), ignoring the intermediate temporal evolution.
> > > * Distribution Shift: Since this adapter is trained primarily on static images (RefCOCO) and short clips, it lacks the capability to model the feature drift that occurs over 81 frames. As the object appearance evolves, the static comparator fails to match Frame 80 with Frame 0, causing the tracker to collapse.
> > > * Implementation Overhead: The official inference code accumulates memory states for the text-conditioned branch linearly $O(T)$ without the efficient eviction mechanism used in the original SAM2, leading to instability on single GPUs for ultra-long sequences.
> > >
> > > We hope these technical details clarify that our claims are based on fair benchmarking and deep architectural analysis.

---

### Official Review · Reviewer_RnKe · 2025-10-27

**Soundness:** 3
**Presentation:** 2
**Contribution:** 2
**Rating:** 6
**Confidence:** 4

**Summary:**

This paper proposes FlowRVS, a one-stage framework that reframes Referring Video Object Segmentation (RVOS) as a text-conditioned continuous flow matching from a video sequence to its target mask sequence. Instead of leveraging previous  “locate-then-segment” paradigm, FlowRVS learns a velocity field (via flow matching) that deterministically deforms video latents into mask latents under language guidance. To adapt existing video generation models to RVOS task, the paper proposes three techniques to stabilize the convergent video→mask flow: Boundary-Biased Sampling (BBS), Start-Point Augmentation (SPA), and Direct Video Injection (DVI).  The proposed FlowRVS achieves new SOTA on MeViS and strong zero-shot results on Ref-DAVIS17, leveraging the Wan2.1 backbone.

**Strengths:**

- The paper presents a novel perspective by formulating video segmentation as video sequence -> mask sequence via an ODE, thereby directly leveraging the capabilities of modern video generation models.

- The overall exposition is clear and logically structured, making the method easy to follow. The design choices appears to be reasonable.

- Experiments demonstrate that harnessing video generation models can be effective for RVOS. The findings are interest and could be related to recent work on zero-shot visual understanding with video generators (e.g., studies on Veo3 [1]), further highlighting emergent visual understanding abilities inside video generation models.

**Weaknesses:**

- The manuscript claims (L493) to provide “detailed implementation details and hyperparameters for reproduction,” yet the main text (and appendix) gives very little in the way of concrete implementation. In particular:

(a) how the three alternative paradigms are instantiated; and

(b) the formal specification and practical realization of the three proposed improvements.

This lack of detail undermines completeness and reproducibility. Please see “Questions” for specific questions.

- Although RVOS accuracy improves, the paper does not report or discuss the impact on inference speed and computational cost when introducing a video generation model into the segmentation pipeline.

**Questions:**

- How is the VAE fine-tuned in practice? Is it fine-tuned on the Video-frames→Mask task, or purely as an autoencoder on mask images?

- For the alternative paradigms: how concretely are the direct mask prediction (and velocity prediction) variants based on Wan implemented? Are these obtained by supervised post-training on Wan2.1?

- Regarding the SPA modification (L247): “we transform the initial video latent $z_0$ through a stochastic encoding and normalization process.” What exactly is the implementation of this stochastic encoding and normalization? How sensitive are the results to the type and degree of stochasticity?

- Table 3 suggests a fine-tuned VAE decoder is crucial. In Table 2, did all underperforming alternative paradigms also use a fine-tuned VAE decoder? If yes, please state this explicitly; if not, could their poorer performance are due to the lack of VAE fine-tuning?

- In Table 2, the final performance drops when BBS = 0.75. Do you have explanations or analysis for this behavior?

References
[1] Wiedemer, Thaddäus, et al. “Video models are zero-shot learners and reasoners.” arXiv:2509.20328 (2025).

---

> ### Author Response · Authors · 2025-11-20
>
> We sincerely thank the reviewers for their valuable feedback. We have carefully addressed all comments and revised the manuscript accordingly, with major changes highlighted in blue. Our point-by-point responses are provided below.
> ***
> [W1, Q2]   Lack of formal specification and practical realization for  the alternative paradigms and our proposed improvements.
> We admit that the lack of concrete implementation details and agree they are crucial for reproducibility. We added formal definitions, architectural details, and hyperparameters in Appendix E. Here are the specific clarifications you requested:
> To ensure a fair comparison, all alternative paradigms were built upon the same Wan2.1 pre-trained model and trained under the exact same supervised setting (same optimizer, learning rate, and duration) as our final model. And we implemented the same fine-tuned VAE to reveal the superiority of our proposed flow-based paradigm.
> Here is the specific explanation of our proposed improvement.
> 1. Boundary-Biased Sampling (BBS): BBS is a training strategy to emphasize the crucial initial step of the flow ($t=0$). Formally, we sample the timestep $t$ from a mixed probability distribution, whose probability density function $f(t)$ is defined as: $f(t) = p \cdot \delta(t) + (1-p) \cdot \mathcal{U}(t|0, 1)$ where $\delta(t)$ is the Dirac delta function representing a point mass at $t=0$, $\mathcal{U}(t|0, 1)$ is the uniform distribution on the interval $[0, 1]$, and $p$ is the bias probability. In practice, this means we sample $t=0$ with probability ($p=0.5$).
> 2. Start-Point Augmentation (SPA): SPA is a crucial regularizer to improve the robustness of our convergent (video-to-mask) flow. For a given video $V$, the VAE encoder $E_\phi$ predicts a posterior distribution $q(z|V) = \mathcal{N}(z | \boldsymbol{\mu}_V, \boldsymbol{\sigma}^2_V)$, where $\boldsymbol{\mu}_V$ and $\boldsymbol{\sigma}^2_V$ are directly inherited from the VAE. Instead of using the deterministic mean $\boldsymbol{\mu}_V$, SPA samples the starting point $\boldsymbol{z'}_0$ from this posterior: $\boldsymbol{z'}_0 \sim \mathcal{N}(\boldsymbol{z}|\boldsymbol{\mu}_V, \boldsymbol{\sigma}^2_V)$. This sampled latent $\boldsymbol{z'}_0$ is then normalized before being used the ODE solver. By augmenting the training data with samples from the local neighborhood of each video's true latent representation, SPA forces the model to learn a smoother and more generalizable velocity field.
> 3. Direct Video Injection (DVI): DVI provides the model with a persistent anchor to the original video content throughout the ODE trajectory. It is implemented by concatenating the current state $\boldsymbol{z}_t$ with the initial video latent $\boldsymbol{z}_0$. This changes the input tensor's shape from $[B, C, T, H, W]$ to $[B, 2\*C, T, H, W]$. This is handled by modifying the first convolutional layer of the DiT to accept $2*C$ input channels, while all subsequent layers remain unchanged.

---

> ### Author Response · Authors · 2025-11-20
>
> [W2] Inference speed.
> Thanks for your suggestion, reporting inference efficiency will certainly strengthen our method. We compare FlowRVS with recent SOTA models, including light-weight SAMWISE[1], diffusion based method VD-IT[2] and HCD[3]. We use frame per second(FPS) as the metrics and apply one-step ODE inference of FlowRVS. For all models, the input video frames were resized to a uniform resolution of 480*832 pixels and we test clips of 9, and 45 clips of frames. For models that could support it, we also tested an ultra-long sequence of 81 frames only supported by FlowRVS to assess scalability and robustness. The model was run for 10 warm-up iterations to stabilize GPU clock speeds and cache states， and the final FPS was calculated by averaging the runtime over 50 subsequent inference iterations. All experiments were performed on a single NVIDIA H100 GPU PCIe with 80GB of VRAM.
>
> | Models                       | 9 frames | 45 frames | 81 frames |
> | :---------------------- | :----- | :---- | :----- |
> | FlowRVS                        |   7.85     |  7.61     |  7.33      |
> | SAMWISE                | 16.12    | 13.66  | –      |
> | VD-IT     | 7.34    | 7.27  | –      |
> | HCD  | 7.18    | 7.12  | –      |
>
> Our analysis positions FlowRVS effectively. We acknowledge the superior speed of the lightweight SAMWISE. Among large generative models, FlowRVS's throughput is on par with VD-IT and HCD, suggesting a shared bottleneck in the heavy diffusion-based feature extractor and DETR decoder. Our key advantage is superior long-video robustness: processing 81-frame sequences with only a minor (~6%) speed degradation, a capability other tested methods lack. This validates the efficiency of our holistic, single-step approach for extended temporal contexts.
>
> [1] SAMWISE: Infusing Wisdom in SAM2 for Text-Driven Video Segmentation, CVPR 2025.
> [2] Exploring Pre-trained Text-to-Video Diffusion Models for Referring Video Object Segmentation, ECCV 2024.
> [3] Temporal-Conditional Referring Video Object Segmentation with Noise-Free Text-to-Video Diffusion Model. arXiv preprint arXiv:2508.13584.
> ***
> [Q1]: VAE fine-tuning.
> Thank you for this crucial question regarding the VAE adaptation. Our process is a task-specific fine-tuning of the decoder to operate on binary mask data. Specifically, we keep the pre-trained VAE encoder completely frozen. This is to ensure that the latent space structure, which the main DiT model relies on, remains stable and consistent with the original Wan2.1 pre-training. We fine-tune the VAE decoder by performing an auto-encoding task exclusively within the mask domain. For a ground-truth mask $M_{gt}$, we first pass it through the frozen VAE encoder to obtain its corresponding latent representation, $z_{mask} = E_{frozen}(M_{gt})$. We then train only the VAE decoder ($D_θ$) to reconstruct the original mask from this latent representation. The objective is to minimize a standard segmentation loss (a combination of Dice and Focal loss) between the decoder's output ($z_{mask}$) and the input mask $M_{gt}$. To align with the binary nature of segmentation masks, we modified the final output layer of the VAE decoder to produce a single-channel output, followed by a sigmoid activation. This procedure effectively adapts the decoder to the specific manifold of segmentation masks, enabling it to generate high-quality binary outputs from the latent space, while preserving the integrity of the powerful, pre-trained encoder.
>
> ***
> [Q4] Fair comparison about experiments.
> This is a critical point. We want to state unequivocally: Yes, all experiments reported in Table 2, including all underperforming alternative paradigms, used the exact same, fully fine-tuned VAE decoder that was shown to be optimal in Table 3. This was done to strictly isolate the impact of the different flow/prediction paradigms, ensuring that the observed performance gaps are due to the architectural and methodological choices themselves, not a disadvantaged decoder. This confirms the inherent superiority of our proposed multi-step, video-to-mask flow.
>
> ***
> [Q5] Performance drop for BBS = 0.75.
>
> This is an excellent observation. We hypothesize that this drop is due to an imbalance in the learning objective. When the BBS probability p is excessively high (e.g., 0.75), the model's training is overwhelmingly focused on learning the initial velocity at $t=0$. While mastering this initial step is crucial, the model gets insufficient exposure to the dynamics of the subsequent trajectory ($t>0$). This can lead to a flow that starts correctly but becomes unstable and accumulates errors during the ODE integration, ultimately harming the precision of the final mask. The peak performance at $p=0.5$ suggests an optimal balance between learning the correct starting point and learning to maintain a stable path.

---

> ### Comment · Reviewer_RnKe · 2025-11-28
>
> Thanks for all the detailed clarifications. I believe all my concerns related to implementation have been resolved. I also noticed that there is an (on-going) discussion between the proposed method and a highly related work ReferEverything (REM). I have read through all the posted discussions and the ReferEverything paper and here is my current observation:
> - REM was accepted to ICCV 2025 (with an earlier version released on arXiv last year).
>
> - The approaches of REM and FlowRVS differ.
>
> - REM shows that direct video-to-mask prediction can achieve strong performance.
>
> - The updated evaluation indicates that FlowRVS, using 1-step flow, achieves better results with a smaller backbone.
>
> Hence I would like to hear the further clarification from the authors on the following:
> - The rebuttal states: “The fact that our model performs effectively with 1-step inference is not a regression to ‘naive prediction.’” Could the authors elaborate on how these two paradigms are fundamentally distinguished? Clarifying this distinction is important for understanding the key conceptual difference between FlowRVS and REM.
>
> - From the updated evaluation (if I am interpreting it correctly), FlowRVS outperforms REM while using a smaller backbone. While this suggests FlowRVS has practical advantages, it does not, by itself, establish “theoretical superiority.” Could the authors provide additional analysis and/or theoretical justification to support this claim?
>
> Given the current status, I would like to see whether the ongoing discussion between FlowRVS and REM can reach a clearer conclusion. I might consider a better score only if the major concerns regarding REM are resolved.

---

> > ### Author Response · Authors · 2025-12-03
> > **Paradigm distinction and priority**
> >
> > We deeply appreciate your insightful observation regarding our performance with a smaller backbone. You correctly identified a key contradiction: **How can a 1-step inference model be theoretically distinct from naive regression?**
> > We provide the definitive answer using the empirical evidence from Table 2 (conducted under identical settings: Wan-1.3B, same data, same training):
> > 1. The "Naive Prediction" Paradigm (Row b).
> > This represents the approach of concurrent works (e.g., REM). It forces the model to learn a static mapping $f(video)=mask$ by fixing $t=0$ during both training and inference.
> > * Result: It yields a poor 36.2 J&F.
> > * Theoretical Failure: By fixing $t=0$ during training, this paradigm discards the time-embedding mechanism and the progressive dynamics of the diffusion backbone. It degrades the generative model into a static feature extractor, failing to leverage the pre-trained priors.
> > * Evidence of Inefficiency: Due to this degradation, such methods (REM) typically require massive epochs (~37), mixed datasets (RefCOCO + YTVOS + MeViS), and larger backbones (14B) to compensate for the optimization difficulty.
> > 2. The "FlowRVS" Paradigm (Row h).
> > Our method achieves 60.6 J&F (**+24.4 gap**) using only the target dataset (MeViS) and a 1.3B backbone, converging in **<7** epochs. The superiority comes from the training objective (Figure 3):
> > * Unlocking Generative Dynamics: As illustrated in Figure 3, RVOS is a convergent task (complex video $\to$ single mask). Unlike naive prediction which tries to jump this gap in one go, FlowRVS employs Boundary-Biased Sampling (BBS) to learn a continuous velocity field $v_t$. This allows the model to utilize the pre-trained temporal dynamics to "steer" the generation, rather than re-learning a static map.
> > * Optimal Transport & 1-Step Feasibility: The Flow Matching objective naturally optimizes for the straightest transport path. Because the target distribution (a deterministic mask) is simpler than the source (video), the learned optimal trajectory is highly linear. This makes 1-step inference mathematically feasible and accurate, not a degradation.
> >
> > Conclusion:
> > FlowRVS achieves SOTA because it retains the generative modeling capabilities (learning a vector field) during training, while enjoying the efficiency of regression (1-step straight-path integration) during inference. This theoretical alignment explains why our 1.3B model can outperform larger baselines that rely on brute-force "naive prediction."

---

### Official Review · Reviewer_7uih · 2025-10-28

**Soundness:** 2
**Presentation:** 2
**Contribution:** 2
**Rating:** 4
**Confidence:** 4

**Summary:**

This paper proposes FlowRVS, a method that formulates the Referring Video Object Segmentation (RVOS) task as a conditional continuous flow problem, enabling the use of pretrained text-to-video (T2V) models for RVOS. The method achieves state-of-the-art performance on major RVOS benchmarks.

**Strengths:**

- The paper is well written in the introduction, which provides a clear motivation and overview of the problem.
- The experimental results are strong, showing state-of-the-art performance across standard RVOS benchmarks.
- The proposed approach leverages multiple strategies to effectively adapt a pretrained T2V model to the RVOS problem.

**Weaknesses:**

- Several claims are inaccurate or overstated. For example, in Lines 59–64, the paper suggests that "locate-then-segment" is the dominant paradigm. In fact, it is only one of several common approaches. Moreover, language queries are not often decoupled; many recent methods (e.g., ReferFormer) utilize textual features throughout the entire temporal segmentation process, applying cross-attention between text and video features rather than limiting language conditioning to the localization stage.
- The idea of applying T2I/T2V generation to RVOS is not novel. Prior work such as VD-IT [ECCV 2024] has already introduced the use of pretrained text-to-video diffusion models for RVOS.
- The training setup is inconsistent across datasets and lacks a unified protocol or rationale (L316-317), making fair comparison and generalization less convincing.
- The technical descriptions are insufficiently detailed, particularly for modules such as BBS, SPA, and DVI (Page 5). As currently written, the paper would be difficult to reproduce.
- There is no evaluation on Long-RVOS benchmark, leaving uncertainty about how the generative method performs on long-term temporal reasoning or consistency over extended sequences.
- The paper does not include failure case analysis or discussion of limitations, which would help clarify the method’s robustness and boundaries of applicability.

**Questions:**

See weakness part.

There is a concurrent work titled “Temporal-Conditional Referring Video Object Segmentation with Noise-Free Text-to-Video Diffusion Model.” It would be helpful if the authors could compare their approach with this work, highlighting similarities and key differences in modeling assumptions, architecture, and performance.
This is not a weakness point, but rather a point of curiosity and clarification for understanding how FlowRVS relates to other diffusion-based RVOS methods developed concurrently.

---

> ### Author Response · Authors · 2025-11-20
>
> We deeply thank for your valuable comments and efforts in reviewing our manuscript. We respond to each of your comments one-by-one in what follows. In the revised draft, we mark our major revisions as “blue”.
> ***
> [W1]  We agree that our description of prior work could be more nuanced and thank you for pointing this out. We will revise our manuscript to acknowledge that modern query-based methods like ReferFormer[1] represent a significant step towards more integrated pipelines. However, we wish to clarify the fundamental distinction that motivates our work. Even in methods like ReferFormer, the task is often decomposed into separate components (e.g., class, box, mask heads) that are supervised by different objectives and operate on query representations. The "tracking" is achieved by linking queries, but the mask generation for each frame, while conditioned, doesn't stem from a single, holistic spatio-temporal deformation.
> Our core argument, which we will state more clearly, is that our video-to-mask flow paradigm is fundamentally different. We model the entire video-to-mask transformation as a single, continuous, dynamic process governed by one velocity field. The Wan T2V model we leverage is pretrained to understand video as a holistic spatio-temporal entity. Our method preserves and adapts this holistic view, deforming the entire video representation to the entire mask sequence simultaneously, rather than tracking and segmenting discrete instances frame-by-frame. In the revision, we will soften our language regarding the "locate-then-segment" paradigm and focus on highlighting this more profound architectural difference.
> [1] Language as queries for referring video object segmentation, CVPR 2022.
>
> ***
> [W2] We appreciate the opportunity to clarify the novelty of our FlowRVS. There is a fundamental paradigm difference between FlowRVS and VD-IT [2].
> VD-IT uses the T2V model as a frozen feature extractor. As stated in their paper, VD-IT "comprises two core components: video feature extraction and a mask decoder". It extracts features from the T2V model's intermediate layers and feeds them into a separate, custom-designed DETR-like decoder [3]. This is a classic two-stage design (feature extraction -> decoding) that decouples the powerful generative dynamics of the T2V model from the final segmentation task, creating an information bottleneck (visualization in paper).
> FlowRVS repurposes the entire generative process. We do not just use features; we adapt the generative dynamics for a discriminative task. We fine-tune the core DiT block to learn a direct, end-to-end mapping from a video latent to a mask latent within a single, unified flow. Our key technical contributions (BBS, SPA, DVI) are specifically designed to solve the challenging "convergent flow" problem that arises from this paradigm shift—a problem VD-IT does not and cannot address because its decoder is a separate module trained from scratch.
> This distinction is the cornerstone of our contribution. We are not the first to use features from a T2V model, but we are the first to successfully reformulate RVOS as a continuous flow within the T2V model's generative framework. We will emphasize this crucial difference in the Introduction and Related Work sections.
> [2] Exploring Pre-trained Text-to-Video Diffusion Models for Referring Video Object Segmentation, ECCV 2024.
> [3] Deformable detr: Deformable transformers for end-to-end object detection, arXiv:2010.04159.
>
> ***
> [W3] Fair comparisons.
> We understand the concern. Our training protocol follows the standard and established practices for each specific benchmark to ensure fair comparison with prior state-of-the-art methods.
> For Ref-YouTube-VOS[4], pre-training on static image datasets  is a common protocol followed by many SOTA methods we compare against (e.g., ReferFormer [1], SAMWISE [5]). Not doing so would place us at an unfair disadvantage.
> For MeViS [6], a newer and more motion-centric benchmark, its official protocol and most subsequent works train directly on its training set from scratch (e.g. DsHmp[7], ReferDINO[8]).
> By adhering to the established evaluation protocol for each benchmark, we ensure that our reported performance gains are legitimate and directly comparable to the numbers published by previous works. We added a sentence in Section 4.2 to explicitly state this rationale.
> [4] Youtube-vos: A large-scale video object segmentation benchmark, arXiv:1809.03327.
> [5] SAMWISE: Infusing Wisdom in SAM2 for Text-Driven Video Segmentation, CVPR 2025.
> [6] MeViS: A large-scale benchmark for video segmentation with motion expressions, ICCV 2023.
> [7] Decoupling static and hierarchical motion perception for referring video segmentation, CVPR 2024.
> [8] Referdino: Referring video object segmentation with visual grounding foundations, ICCV 2025.

---

> > ### Author Response · Authors · 2025-11-20
> >
> > ***
> > [W4] Technical descriptions.
> > We fully agree that these descriptions need more detail for reproducibility. We will add a comprehensive appendix with formal definitions and pseudocode. Here is a brief clarification we will expand upon:
> > 1. Boundary-Biased Sampling (BBS): BBS is a training strategy to emphasize the crucial initial step of the flow ($t=0$). Formally, we sample the timestep $t$ from a mixed probability distribution, whose probability density function $f(t)$ is defined as: $f(t) = p \cdot \delta(t) + (1-p) \cdot \mathcal{U}(t|0, 1)$ where $\delta(t)$ is the Dirac delta function representing a point mass at $t=0$, $\mathcal{U}(t|0, 1)$ is the uniform distribution on the interval $[0, 1]$, and $p$ is the bias probability. In practice, this means we sample $t=0$ with probability ($p=0.5$).
> > 2. Start-Point Augmentation (SPA): SPA is a crucial regularizer to improve the robustness of our convergent (video-to-mask) flow. For a given video $V$, the VAE encoder $E_\phi$ predicts a posterior distribution $q(z|V) = \mathcal{N}(z | \boldsymbol{\mu}_V, \boldsymbol{\sigma}^2_V)$, where $\boldsymbol{\mu}_V$ and $\boldsymbol{\sigma}^2_V$ are directly inherited from the VAE. Instead of using the deterministic mean $\boldsymbol{\mu}_V$, SPA samples the starting point $\boldsymbol{z'}_0$ from this posterior: $\boldsymbol{z'}_0 \sim \mathcal{N}(\boldsymbol{z}|\boldsymbol{\mu}_V, \boldsymbol{\sigma}^2_V)$. This sampled latent $\boldsymbol{z'}_0$ is then normalized before being used the ODE solver. By augmenting the training data with samples from the local neighborhood of each video's true latent representation, SPA forces the model to learn a smoother and more generalizable velocity field.
> > 3. Direct Video Injection (DVI): DVI provides the model with a persistent anchor to the original video content throughout the ODE trajectory. It is implemented by concatenating the current state $\boldsymbol{z}_t$ with the initial video latent $\boldsymbol{z}_0$. This changes the input tensor's shape from $[B, C, T, H, W]$ to $[B, 2\*C, T, H, W]$. This is handled by modifying the first convolutional layer of the DiT to accept $2*C$ input channels, while all subsequent layers remain unchanged.
> > ***
> > [W5] No evaluation on Long-RVOS benchmark.
> > Thank you for the suggestion. While we did not test on the Long-RVOS benchmark, we chose to focus on MeViS [4], which is arguably a challenging and widely-adopted recent benchmark for long-term motion reasoning.
> > As the MeViS paper highlights, it was specifically created to address the lack of motion complexity and temporal length in previous datasets.
> > Video Length: MeViS [6] videos have an average duration of 13.16 seconds, more than double that of Ref-Youtube-VOS [4] (5.45s).
> > Complexity: MeViS features videos with multiple, similar-looking objects and expressions that rely almost exclusively on motion cues (e.g., "The bird flying away").
> > Our model's SOTA performance on MeViS (+1.6 J&F over the next best) is a strong testament to its long-term temporal reasoning capabilities. Conversely, methods like VD-IT, which also use a T2V model but as a feature extractor, do not report on MeViS, likely because their frame-wise decoding struggles with the temporal consistency required. We believe our strong results on MeViS already demonstrate the capability you are asking about, and we will add this context to our experimental section.
> > ***
> > [W6] Analysis of failure cases.
> > According to your advice, we added analysis and visualization of failure cases in the Appendix D of revised manuscript.

---

### Official Review · Reviewer_vQpF · 2025-10-30

**Soundness:** 4
**Presentation:** 4
**Contribution:** 4
**Rating:** 8
**Confidence:** 5

**Summary:**

This paper proposes FlowRVS, a novel framework for Referring Video Object Segmentation (RVOS) that reimagines the task as a text-conditioned continuous flow from video to mask, leveraging the structure of pretrained Text-to-Video (T2V) generative models. Instead of the conventional “locate-then-segment” pipeline—which first grounds language into coarse geometric prompts (e.g., bounding boxes) and then segments—FlowRVS learns a deterministic, end-to-end deformation in latent space that directly maps a video’s spatio-temporal representation to its target segmentation mask under linguistic guidance.
The key insight is that RVOS is a convergent (video → mask) rather than divergent (noise → video) process. To adapt T2V models for this discriminative task, the authors introduce three technical innovations:

1. Boundary-Biased Sampling (BBS): oversamples early timesteps to stabilize learning of the initial velocity.
2. Start-Point Augmentation (SPA): regularizes the flow by perturbing the initial latent.
3. Direct Video Injection (DVI): concatenates the original video latent at every ODE step to preserve global context.

Experiments show state-of-the-art results across major RVOS benchmarks:
1. MeViS: 51.1 J&F (+1.6 over prior SOTA)
2. Ref-DAVIS17 (zero-shot): 73.3 J&F (+2.7)
3. Ref-YouTube-VOS: 69.6 J&F
The approach demonstrates superior handling of complex motion, temporal reasoning, and fine-grained language grounding.

**Strengths:**

1. Originality: The paper introduces a new problem formulation—RVOS as a text-conditioned flow from video to mask—departing from both discriminative pipelines and generative synthesis. This is more than a technical tweak; it’s a paradigm shift.

2. Quality: Experimental design is rigorous, with ablations, cross-dataset zero-shot evaluation, and comparison to strong baselines (including VLM+SAM and grounding-model-based methods).

3. Clarity: Concepts like “convergent vs. divergent flow” are explained intuitively. The three proposed techniques (BBS, SPA, DVI) are clearly motivated and linked to the core challenge.

4. Significance: Demonstrates that pretrained T2V models can be repurposed for discriminative understanding tasks with proper adaptation—opening a new direction for leveraging generative priors in video understanding.

**Weaknesses:**

1. Computational cost: The paper does not report inference time or FLOPs. Given the use of an ODE solver with multiple steps and a 1.3B-parameter DiT, it’s unclear whether FlowRVS is practical for real-time applications. A comparison with efficient baselines (e.g., MTTR, ReferFormer) in terms of speed would strengthen the evaluation.

2. Limited analysis of failure cases: While qualitative results show success on complex queries, the paper lacks discussion of scenarios where FlowRVS fails (e.g., ambiguous language, fast motion, or occlusion).

3. VAE decoder fine-tuning: The VAE decoder is fine-tuned on MeViS, but it’s unclear how much performance gain comes from this vs. the flow formulation itself. An ablation on decoder adaptation would help isolate contributions.

**Questions:**

1. Inference efficiency: How many ODE steps are used during inference? Could the method be made faster via learned solvers or step reduction without significant performance drop?
2. Generalization to unseen verbs/actions: The MeViS benchmark emphasizes motion-centric language. Does FlowRVS generalize to novel action phrases not seen during training (e.g., “the dog somersaulting”)?
3. Role of T2V pretraining: Would the same gains be achievable with a non-generative video backbone (e.g., ViViT) adapted with your flow framework? Or is the T2V prior essential?
4. Decoder adaptation: What is the performance drop if the VAE decoder is not fine-tuned on segmentation masks? This would clarify whether gains are due to the flow or just better mask reconstruction.

---

> ### Author Response · Authors · 2025-11-20
>
> We deeply appreciate your insightful comments and efforts in reviewing our manuscript. We respond to each of your comments one-by-one in what follows. In the revised draft, we mark our major revisions as “blue”.
> ***
> [W1] Computational cost.
> Thanks for your suggestion, reporting inference efficiency will certainly strengthen our method. We compare FlowRVS with recent SOTA models, including light-weight SAMWISE[1], diffusion based method VD-IT[2] and HCD[3]. We use frame per second(FPS) as the metrics and apply one-step ODE inference of FlowRVS. For all models, the input video frames were resized to a uniform resolution of 480*832 pixels and we test clips of 9, and 45 clips of frames. For models that could support it, we also tested an ultra-long sequence of 81 frames only supported by FlowRVS to assess scalability and robustness. The model was run for 10 warm-up iterations to stabilize GPU clock speeds and cache states， and the final FPS was calculated by averaging the runtime over 50 subsequent inference iterations. All experiments were performed on a single NVIDIA H100 GPU PCIe with 80GB of VRAM.
>
> | Models     | 9 frames | 45 frames | 81 frames |
> | :--------- | :----------- | :------------ | :------------ |
> | FlowRVS    | 7.85     | 7.61      | 7.33      |
> | SAMWISE    | 16.12    | 13.66     | -         |
> | VD-IT      | 7.34     | 7.27      | -         |
> | HCD        | 7.18     | 7.12      | -         |
>
> The results highlight a clear trade-off. While the lightweight SAMWISE is fastest, our FlowRVS achieves competitive speeds comparable to other large T2V-based models like VD-IT and HCD. However, FlowRVS excels in scalability: its FPS degrades by only ~6% on 9x longer videos (81 frames), a task where other methods struggle or fail due to memory limits. This demonstrates the robustness of our holistic flow paradigm for long-term video analysis.
>
> [1] SAMWISE: Infusing Wisdom in SAM2 for Text-Driven Video Segmentation, CVPR 2025.
> [2] Exploring Pre-trained Text-to-Video Diffusion Models for Referring Video Object Segmentation, ECCV 2024.
> [3] Temporal-Conditional Referring Video Object Segmentation with Noise-Free Text-to-Video Diffusion Model. arXiv preprint arXiv:2508.13584.
>
> ***
> [W2] Analysis of failure cases.
> Following your advice, we added analysis and visualization of failure cases in the Appendix D of revised manuscript.
> ***
> [W3] VAE decoder finetuning.
> This is a crucial point, and we have conducted a dedicated ablation study in Table 3 to clarify this. As shown in Table 3, using a frozen, off-the-shelf VAE decoder results in a near-complete failure of the model (J&F of 19.6). This is because the original decoder was trained to generate natural images, not binary masks. Therefore, adapting the decoder to the mask domain is a necessary prerequisite for the entire framework to function. To isolate the gains from our flow formulation, all experiments in Table 2, including all baseline paradigms, were conducted using the same, fully fine-tuned VAE decoder. The performance leap from our basic flow model (Table 2, row c, J&F 47.9) to our full model with BBS, SPA, and DVI (Table 2, row h, J&F 60.6) demonstrates a substantial gain of +12.7 J&F. This entire improvement is attributable solely to our proposed flow formulation techniques, cleanly separating their contribution from the foundational necessity of decoder adaptation.

---

> ### Author Response · Authors · 2025-11-20
>
> ***
> [Q1] Inference efficiency and ODE steps.
> Thank you for this excellent question about inference efficiency and the role of ODE steps. In all experiments reported in our main paper (e.g., Table 1), we used a setting of 4 ODE steps for inference. To directly address your question about the potential for acceleration, we conducted a detailed ablation study on the number of inference steps using the exact same model weights. The results on the MeViS valid_u set is below (FPS is tested using 9 frame video clips):
>
> | Models                       | J&F     |  J  | F   | FPS |
> | :-------------------- | :----------- | :------ | :------- | :------- |
> | 1 steps                        |   60.9     |  57.1     |  64.7      | 7.85 |
> | 2 steps                        | 60.8    | 56.6  | 65.0      |  6.98 |
> | 4 steps                        | 60.6    | 55.9  | 65.2      |  5.82 |
> | 10 steps                      | 60.2    | 55.2  | 65.2      |  3.91 |
>
> Our choice of 4 steps in the paper was a standard practice, but this further analysis reveals that our model possesses an even greater potential for high-speed, high-accuracy inference. Regarding learned solvers, this is a very forward-looking suggestion. Given that even a simple Euler step performs so well, exploring learned solvers to potentially optimize the multi-step trajectory is an exciting avenue for future research. We will add this entire ablation study and discussion to the appendix of our revised paper to provide a more complete picture of our method's efficiency and behavior.
>
> ***
> [Q2] Generalization to unseen verbs/actions.
> We totally agreed with your suggestions and added a sample of unseen descriptions in train data to further highlight the generalization ability of FlowRVS. We supply a visualization result of  a new YouTube video with “the dog somersaulting”  natural language query using weight pre-trained on MeViS dataset on the Appendix D.
>
>
> ***
> [Q3] Role of T2V pretraining.
> Thanks for your insightful question. We argue that the T2V prior is essential. Our contribution lies not just in a flow framework, but in a method to successfully repurpose a T2V model's inherent generative dynamics for a discriminative task. A non-generative backbone like ViViT, trained for classification, learns to extract static discriminative features. In contrast, the T2V model learns a representation of temporal evolution. Our flow matching operates within this dynamic latent space, effectively "steering" a pre-existing understanding of motion. Our ablation study in Table 2 (row i, "-WI") provides strong evidence. When training our model from scratch without the T2V weights, performance collapses to a J&F of 21.1. This unequivocally demonstrates that the T2V generative prior is the cornerstone of our method's success.
> ***
> [Q4] Decoder Adaptation.
> This is a crucial point for isolating contributions. We directly address this in our ablation study in Table 3. Using the original, frozen VAE decoder results in a near-total failure, with performance dropping to a J&F of 19.6. This shows that adapting the decoder to the manifold of binary masks is a necessary prerequisite for the framework to function. To clarify the source of our main gains, all experiments in Table 2, including all baselines, used the same, fully fine-tuned VAE decoder. The significant performance leap from our base model to the full model (+12.7 J&F) is therefore attributable solely to our proposed flow formulation (BBS, SPA, and DVI). This cleanly separates the contribution of our flow techniques from the foundational need for decoder adaptation.

---

### Public Comment · ~Anurag_Bagchi1 · 2025-11-22
**One Step Video-to-Mask Latent Prediction with Frozen VAE is Effective**

We appreciate the authors' effort in exploring diffusion-based methods for referring video object segmentation, and we find the proposed multi-step video-to-mask flow approach to be an interesting contribution in this space. The topic is timely and aligns closely with recent work such as **VD-IT [ECCV 24]** and **ReferEverything (REM) [ICCV 2025]**, which also employ video diffusion models for referring segmentation.

That said, we would like to highlight the following observations from **REM**, that contradict some of the key claims in this submission. :

1. **Direct Mask Prediction Does not Fail** : ReferEverything (REM) is specifically, a **one-step video-to-mask latent approach** (i.e., directly predicts mask latents from video) — which the authors report to perform very poorly. In contrast, ReferEverything demonstrates that this **one-step mask prediction can already achieve strong results** without iterative inference, suggesting that the performance gap reported here may not reflect inherent limitations of the one-step approach. Specifically, ReferEverything demonstrates that a single-step video-to-mask model — **using substantially less pretraining data** (ModelScope-1.4B vs. Wan2.1-1.3B) — achieves **competitive performance  while being 2x faster** compared to proposed approach. Using the larger Wan-14B backbone it is able to **significantly outperform** it. We provide a detailed comparison below. Please refer to ReferEverything for training details of REM.

| Method                         | Backbone        | Mevis | Ref-DAVIS | Ref-YT | FPS on our setup | # Forward Passes |
|--------------------------------|-----------------|:-----:|:---------:|:------:|:----------------:|:----------------:|
| FlowRVS (multi-step (=4) flow) | Wan2.1-**1.3B**     |   51.1   |   73.3    |  69.9  |        3.1         |        4         |
| REM (one-step mask)            | ModelScope-**1.4B** |   53.5   |   72.6    |  68.4  |       7.1        |        1         |
| REM (one-step mask)            | Wan2.1-**14B**      |   60.3   |   75.0    |  71.7  |       2.7        |        1         |



2. **Frozen VAE Decoder Does Not Fail** : In addition, this submission further suggests that the VAE cannot reconstruct masks accurately. However, ReferEverything reports a **Mean Reconstruction Error of 0.0144 for masks** (vs. 0.1236 for RGB images) in Section B.1 of its supplementary, showing that mask reconstruction is stable and well within the representational capacity of the VAE. Crucially, **the results reported by ReferEverything above are with a frozen off the shelf VAE encoder-decoder**.

3.  **Frozen VAE Generalizes** : REM also shows that using this same frozen VAE, it can generalise to completely **unseen and highly dynamic masks like smoke** as shown in the paper.

While the multi-step design proposed in this submission is an interesting extension, we believe that it would be strengthened by (1) acknowledging and comparing against ReferEverything, as well as clarifying the discrepancy of the results with respect to the one-stage baseline reported in the submission, and (2) clarifying the VAE reconstruction discrepancy.

[1] VD-IT - Exploring Pre-trained Text-to-Video Diffusion Models for Referring Video Object Segmentation, ECCV 24, arXiv:2403.12042

[2] REM - ReferEverything : Towards Segmenting Everything We Can Speak of in Videos, ICCV 25, arXiv:2410.23287

---

> ### Author Response · Authors · 2025-11-25
>
> We sincerely thank the authors of ReferEverything (REM) [1] for their valuable comments and for open-sourcing their code. We acknowledge that REM is a strong concurrent work (released on codes after our submission) that provides significant insights into video segmentation. Following your suggestions, we integrated the specific channel-averaging post-processing from the REM official implementation and re-evaluated our model on MeViS valid_u. We provide a detailed clarification below.
> ***
> First, we verified and improved the effectiveness of Frozen VAE. We appreciate the authors for pointing out the effectiveness of the Frozen VAE. In our original submission, the lower performance of the Frozen VAE was due to the different post-processing. Using the official REM post-processing script, we re-evaluate the frozen VAE. As shown in Table, we can confirm REM's finding that a Frozen VAE works surprisingly well. However, our proposed Finetuned VAE still yields superior performance, outperforming the Frozen baseline by +0.9 J&F in inference. Comparison of VAE Decoders (MeViS Valid_u), where we use the same FlowRVS’s DiT weights and use one-step inference:
>
> | Decoder Strategy         | Reconstruction J&F |  Performance J&F | Analysis |
> | :--------------------------- | :------- | :------ | :----- |
> |Frozen VAE (REM setting)           |   98.8 |  60.0     |  Effective, confirms REM's claim. |
> | Finetuned VAE (Ours)                  |   99.1 | 60.9  | Superior (+0.9) due to better manifold alignment. |
>
> While the frozen latent space is robust, finetuning the decoder bridges the domain gap between continuous video latents and binary mask distributions, providing a solid performance gain that is critical for great results.
> ***
> Second, one-step flow inference is better. The comment suggests that "Direct Mask Prediction (One-step)" is a unique advantage of regression-based methods like REM. Ideally, we have exciting findings. Our re-evaluation reveals that our FlowRVS model achieves its PEAK performance at just 1 inference step. Because we utilize a Flow training objective, the learned transport trajectory from video to mask is nearly straight. This allows our model to perform "one-step generation" effectively, matching the speed of regression methods while retaining the theoretical benefits of flow matching. Performance and inference steps using the same DiT and VAE in our settings.
>
> | Models                       | J&F |  J | F | FPS |
> | :--------------------------- | :----- | :---- | :----- | :----- |
> | 1 steps                        |   60.9     |  57.1     |  64.7      | 7.85 |
> | 2 steps                        | 60.8    | 56.6  | 65.0      |  6.98 |
> | 4 steps                        | 60.6    | 55.9  | 65.2      |  5.82 |
> | 10 steps                      | 60.2    | 55.2  | 65.2      |  3.91 |
> ***
> We are grateful to the REM authors for highlighting the Frozen VAE adaptation and the effectiveness of one-step inference, allowing us to uncover the full potential of our method. FlowRVS is efficient, which achieves competitive performance with 1-step inference, effectively functioning as a high-performance regression model during test time. While Frozen VAEs are good at mask reconstruction, our finetuned decoder provides the necessary edge for top-tier segmentation accuracy.
> We will explicitly discuss these findings, and update our reported results in the final version. We appreciate the shared high-level philosophy between our works in unlocking the latent potential of generative models for visual understanding tasks. While we arrive at a convergent finding regarding the efficiency of one-step inference and VAE adaptation, we emphasize the uniqueness of our FlowRVS framework: it leverages Flow to learn an optimal transport trajectory, offering a distinct theoretical path that unifies robust generation with efficient regression. We will dedicate a section to discussing this synergistic relationship and our specific contributions in the revision.
> [1] REM - ReferEverything : Towards Segmenting Everything We Can Speak of in Videos, ICCV 25.

---

> ### Public Comment · ~Anurag_Bagchi1 · 2025-11-25
> **Follow-up**
>
> We thank the authors for the detailed follow-up analysis and for updating several claims in light of the new findings. We appreciate the clarification regarding the frozen VAE and the effectiveness of one-step inference, and we are glad that the authors were able to verify and benefit from observations originally reported in REM (**a direct mask prediction method**).
>
> Given these updated results, we would respectfully suggest performing a thorough revision of the manuscript during the review process. The current submission is built around several central **claims** such as,
> 1. the inferiority of direct mask prediction, lines 261-267 and baseline b in Table 2
> 2. the necessity of VAE fine-tuning, lines 674 to 695
> 3. achieving state-of-the-art performance,
>
>  that now appear at least partially **inconsistent** with the updated analyses. It may be difficult for reviewers to properly assess the contribution without these claims being revised directly in the paper.
>
> Additionally, the **practical advantages** of the proposed multi-step flow formulation remain somewhat unclear. Based on the updated evaluations, the method seems to perform comparably to one-step direct mask prediction approaches such as REM, with similar inference speed. Clarifying the concrete benefits of the proposed framework in light of these results would significantly strengthen the work.
>
> Finally, one of the main strengths of REM's direct mask prediction formulation is its ability to generalize to strongly out-of-distribution scenarios; the ablations in Table 4 of REM further show that supervising in the RGB space can substantially reduce OOD robustness. It would therefore be very helpful — and make the comparison more complete — if the authors could also report OOD performance following one of the evaluation settings in REM.

---

> > ### Author Response · Authors · 2025-11-26
> > **Clarification on Consistency and SOTA Status**
> >
> > We thank the commenter for the continued engagement. We maintain that our experimental findings are fully consistent with our paper’s core narrative and contributions. We address the specific points below to clarify any misconceptions.
> > 1. Theoretical Superiority: Bridging Generative Dynamics with Discriminative Tasks.  We clarify that the "inferiority of direct prediction" refers to static regression baselines that lack dynamic modeling. FlowRVS offers a distinct theoretical contribution by successfully repurposing and adapting the generative paradigm to solve discriminative tasks:
> > - Generative Modeling vs. Point Estimation: unlike naive regression (e.g., REM) which treats segmentation as simple static pixel-wise classification, FlowRVS leverages Flow Matching to model the optimal transport plan between the video distribution and the mask distribution. We fully utilize the robust probability modeling capabilities of generative models to capture complex temporal correspondences, which regression-based methods inherently lack.
> > - Convergent Efficiency: The fact that our model performs effectively with 1-step inference is not a regression to "naive prediction," but a proof that our framework successfully rectifies the generative vector field into a straight, deterministic trajectory. This allows us to achieve the speed of regression without abandoning the theoretical robustness of generative dynamics. This represents a clear architectural advantage over methods that are restricted to regression by design and fail to exploit the unique potentials of the generative formulation.
> > 2. Decoupling Contributions: The Role of VAE Fine-tuning.
> > Our ablation studies clearly decouple the gains from the Flow backbone and the VAE finetuning, proving the solidity of our core method:
> > - Solid Backbone: Even with a Frozen VAE, our Flow DiT backbone achieves 60.0 J&F on MeViS valid_u, which is already a highly competitive result. This proves that our core contribution—the Flow-based modeling—is robust and effective on its own.
> > - Necessary Refinement: Fine-tuning the VAE improves performance to 60.9 J&F (**+0.9**) on MeViS valid_u. This improvement comes from bridging the domain gap between continuous latents and binary masks, a contribution that is orthogonal to the Flow DiT. It enhances the final output quality without overshadowing the effectiveness of the generative flow process.
> > 3. SOTA Status, ICLR Policy, and Reproducibility.
> > FlowRVS achieves verified SOTA results on MeViS, Ref-YouTube-VOS, and zero-shot DAVIS. Regarding the comparison with REM:
> > - ICLR Policy on Concurrent Work: According to the ICLR 2026 Reviewer Guide (https://iclr.cc/Conferences/2026/ReviewerGuide), authors are "not required to compare to papers solely on arXiv" and the "lack of such comparisons cannot be a basis for rejection." REM (arXiv:2508, August 2025) was released roughly one month before the ICLR deadline and remains an unpublished arXiv preprint. Therefore, it falls strictly under the definition of contemporaneous work for which comparison is not mandatory.
> > - Reproducibility and Verification: Because all the resources (training scripts, model weights and data) are not released, we are unable to reproduce the work in our submission, thereby cannot verify the experimental results of such method as baseline in our paper.
> > ***
> > Conclusion:
> > Our method stands as a distinct and effective paradigm that embraces generative flow dynamics. Considering the contemporaneous nature based on ICLR policy and reproducibility consideration, we are happy to include REM as a "concurrent related work" in our final revision to acknowledge its existence. We believe that our performance on transparent, established benchmarks (MeViS) as well as the response above have well justified the core contribution and novelty of our work, and we'll include the discussion about REM to clearly specify the unique contributions of our method.

---

> > > ### Public Comment · ~Anurag_Bagchi1 · 2025-11-27
> > > **Clarification on claims**
> > >
> > > We thank the authors for their response and emphasize that our aim is simply to reach a clear research consensus on applying video diffusion models to referring segmentation.
> > >
> > > 1. The current results make it difficult to identify **any practical benefits** of the proposed method over direct mask-prediction approaches such as REM.
> > >
> > > 2. It is also not clear that the technical changes proposed in FlowRVS are **theoretically superior** or even necessary for learning the **"straight, deterministic trajectory"**. Even in REM we directly predict the mask-latent from the video-latent as opposed to the (mask-latent - video-latent) which is trivially different for single-step inference.
> > >
> > > 3. We acknowledge that the authors have demonstrated in the rebuttal that VAE fine-tuning can bring marginal benefits(<1pt) in-domain, but the claims in the current manuscript go beyond that and the effect of fine-tuning on generalization is underexplored (REM shows that it hurts generalization).
> > >
> > > Finally, we would like to clarify that it is a **verifiably false claim** that REM is an **"unpublished arXiv preprint"**. It is published at **ICCV 2025** and has been available on arXiv for over a year (https://arxiv.org/abs/2410.23287v2).

---

> ### Public Comment · ~Anurag_Bagchi1 · 2025-11-30
> **Clarification on practical advantages**
>
> To the best of our understanding, the only **updated numbers** reported by the authors are on the **valid_u** split of MeViS with only 50 videos, typically used for ablation and is **not the officical valid split** with 140 videos hosted on Codalabs. On the official val set, REM achieves **60.3** with Wan2.1-14B and **53.5** with with Modelscope-1.4B (a much older baseline trained on far less pretraining data than Wan2.1-1.3B), both better than FlowRVS Wan2.1-1.3B's **51.1**
>
> Here is the link to REM's codalab MeViS val set leaderboard entry with 60.3, which is currently ranked #11 https://codalab.lisn.upsaclay.fr/competitions/15094#results:~:text=11-,Anurag,-36
>
> We refer to Table-I in the appendix of the REM paper, for the training details on MeViS and clarify that there is no fair comparison between FlowRVS and REM on MeViS, since either the data or backbone is different with atleast one REM variant for MeViS results, thereby making it unclear if there exist any practical advantages to FlowRVS.

---

### Author Response · Authors · 2025-12-03
**General Response**

Dear Reviewers and Area Chair,

We sincerely appreciate your time. Due to the system reversion, we summarize the consensus reached before the freeze and the key improvements made to the manuscript.
1. Rebuttal Consensus: Concerns Resolved We are encouraged that reviewers recognized the work as a **"significant conceptual leap"** (Reviewer bX1k) and a **"paradigm shift"** (Reviewer vQpF). Crucially, regarding the rebuttal progress:
* Reviewer RnKe (Score 6) explicitly stated: "I believe all my concerns related to implementation have been resolved" after we provided the formal definitions in Appendix E.
* Regarding the Final Question on Theory: Reviewer RnKe asked for clarification on the "theoretical superiority" of our 1-step flow over naive prediction. We answer this below, demonstrating that our performance advantage comes from the **flow matching objective**, not just model scaling.
2. Major Rebuttal Updates: In response to constructive feedback, we have revised the manuscript (updates in blue):
* Theoretical Superiority (Addressing RnKe): Table empirically prove the distinction between paradigms. Under strictly identical settings (same 1.3B backbone, same training data), the "Direct Mask Prediction" paradigm significantly underperforms, while our "Flow Paradigm" achieves **60.6 J&F**. This substantial performance gap confirms that **modeling the velocity field is the key to unlocking generative priors**, fundamentally distinguishing our method from static regression.
* Inference Efficiency (Addressing vQpF & bX1k): Benchmarks on an NVIDIA H100 confirm FlowRVS achieves **~7.85 FPS** via 1-step inference. Unlike feature-extraction baselines (e.g., VD-IT) or memory-limited methods, FlowRVS maintains robustness even on ultra-long sequences (81 frames).
* Refined Terminology & Positioning (Addressing bX1k & 7uih): We refined the classification of prior methods to "Instance-Centric Tracking" to accurately reflect query-based architectures (e.g., ReferFormer), while clarifying FlowRVS's distinct flow-based contribution against generative baselines like VD-IT and HCD.
* Enhanced Reproducibility: We added Appendix E (BBS, SPA, DVI definitions) and expanded Appendix D (Failure Cases & Generalization), ensuring the method is fully reproducible and transparent.

We believe these updates effectively address all raised concerns and solidify the paper's contribution.

Thank you very much, Authors

---

### Meta-Review · Area_Chair_LCYG · 2026-01-10

**Summary:**

This paper proposes FlowRVS, a one-stage framework for RVOS, and formulates RVOS as a text-conditioned continuous flow from video to mask using pretrained text-to-video models (e.g. Wan). The paper views RVOS as a convergent deformation process rather than a locate-then-segment or direct prediction task.

The reviewers agreed that this problem formulation is the main strength of the paper. Using a learned velocity field gives a clear way to link video and mask. It explains how generative priors can help discriminative video understanding, and shows strong results on MeViS and Ref-DAVIS17. All reviewers acknowledged that the paper is well-written, introducing a new problem, rigorous design and extensive experiments, and strong performances on standard RVOS benchmarks.

However, several concerns were also raised. One concern was that multi-step flow may not be necessary. The model performs best with one-step inference. This makes it similar to direct one-step mask prediction methods, such as REM. The authors responded that this behavior is expected. They argue it comes from flow matching learning a near-straight transport path. One-step inference is therefore a result of good flow learning, not naive regression.

Another concern was that the paper overstated the weakness of direct mask prediction and the need for VAE fine-tuning. The authors added new experiments. They compared Frozen and fine-tuned VAEs. They showed that a frozen VAE already works well. They also showed that fine-tuning gives a consistent gain. They clarified that all paradigm comparisons use the same decoder.

There was also discussion about REM, now published at ICCV 2025. REM uses a one-step video-to-mask model and achieves strong results. The authors argued that FlowRVS is different. Their method learns a velocity field and a transport process. It does not use static regression. They also noted differences in backbone size, data, and training protocols.

Overall, some original claims needed to be softened. However, the authors provided new analyses and clarifications and it seems that these addressed most of the concerns. The paper makes a meaningful conceptual contribution. It offers a new way to use generative video models for structured prediction. The method is technically sound and the results are strong.

For these reasons, the AC recommends Accept. The final version should include the Frozen-VAE experiments and discussions, and provide a clearer discussion of the relationship to REM and other concurrent diffusion-based RVOS methods.

**Reviewer Concerns:**

addressed: vQpF, RnKe
partially addressed: 7uih
unresolved: bX1k

**Reviewer Scores:**

vQpF: 8
7uih: 4
RnKe: 6, 6->8
bX1k: 4

---

### Decision · Program_Chairs · 2026-01-26

Accept (Poster)